# Interleukin 31 receptor α promotes smooth muscle cell contraction and airway hyperresponsiveness in asthma

Santhoshi V. Akkenepally [1,2,5], Dan J. K. Yombo [3,5], Sanjana Yerubandi [1,3], Geereddy Bhanuprakash Reddy [2], Deepak A. Deshpande [4], Francis X. McCormack[1] & Satish K. Madala [1] ✉

Asthma is a chronic inflammatory airway disease characterized by airway hyperresponsiveness (AHR), inflammation, and goblet cell hyperplasia. Multiple cytokines, including IFNγ, IL-4, and IL-13 are associated with asthma; however, the mechanisms underlying the effects of these cytokines remain unclear. Here, we report a significant increase in the expression of IL-31RA, but not its cognate ligand IL-31, in mouse models of allergic asthma. In support of this, IFNγ, IL-4, and IL-13 upregulated IL-31RA but not IL-31 in both human and mice primary airway smooth muscle cells (ASMC) isolated from the airways of murine and human lungs. Importantly, the loss of IL-31RA attenuated AHR but had no effect on inflammation and goblet cell hyperplasia in mice challenged with allergens or treated with IL-13 or IFNγ. We show that IL-31RA functions as a positive regulator of muscarinic acetylcholine receptor 3 expression, augmenting calcium levels and myosin light chain phosphorylation in human and murine ASMC. These findings identify a role for IL-31RA in AHR that is distinct from airway inflammation and goblet cell hyperplasia in asthma.

Asthma is a common chronic airway inflammatory disease that affects more than 300 million people worldwide. Asthma is associated with significant health and economic burdens and impairs both quality of life and productivity[1,2]. Despite advances in understanding the disease physiopathology and the development of anti-inflammatory therapeutic interventions, the incidence of asthma, including uncontrolled and severe forms, continues to increase. The disease is characterized by episodes of airflow obstruction that are associated with high mortality when severe and unreversed[3]. The primary clinical manifestations in the pathophysiology of chronic asthma include airway hyperresponsiveness (AHR), inflammation, goblet cell hyperplasia with mucus production, and fibrotic airway remodeling[4]. While a cardinal feature of asthma, AHR is also present in other airway disorders

such as cystic fibrosis and COPD[5,6]. Th2 cytokines, IL-4 and IL-13, which are elevated in the lungs and serum of asthmatic patients, are now well-established drivers of AHR and airway inflammation in asthma[7–9]. IL-13 is a major cytokine responsible for goblet cell hyperplasia, tissue remodeling, and bronchospasm. Even in the absence of an allergen exposure or inflammation, IL-13 alone is sufficient to induce AHR and airway inflammation in mice[10,11]. IL-4 is crucial in the regulation of Th2 cell proliferation as well as the antibody class switch to produce IgE, and recent studies highlight a shared role of this cytokine with IL-13 in the induction of AHR and inflammation through their common receptor type II IL-4Rα, as well as through type I IL4Rα[12,13]. Clinical trials that neutralize the common IL-4Rα receptor for these Th2 cytokines have shown improvements in AHR and inflammation in both allergic

[1]Division of Pulmonary, Critical Care and Sleep Medicine, Department of Internal Medicine, University of Cincinnati, Cincinnati, OH, USA. [2]Division of Biochemistry, National Institute of Nutrition, Hyderabad, Telangana, India. [3]Division of Pulmonary Medicine, Department of Pediatrics, Cincinnati Children's Hospital Medical Center, Cincinnati, OH, USA. [4]Division of Pulmonary, Allergy, and Critical Care Medicine, Center for Translational Medicine, Jane and Leonard Korman Respiratory Institute, Sidney Kimmel Medical College, Thomas Jefferson University, Philadelphia, PA, USA. [5]These authors contributed equally: Santhoshi V. Akkenepally, Dan J. K. Yombo. ✉e-mail: madalash@ucmail.uc.edu

asthma and atopic dermatitis patients[14,15]. However, these therapies and other more conventional treatments for asthma have not been effective against low type 2 or non-type 2 endotypes and severe asthma with a mixed Th1/Th2 cytokine phenotype[16]. A better understanding of the mechanisms that regulate AHR and therapeutic resistance in these specific conditions may help to develop alternative therapeutic approaches to control more atypical asthma endotypes.

IL-31 is a Th2-related cytokine implicated in tissue remodeling in chronic lung and skin diseases[17,18], including allergic asthma, atopic dermatitis (AD), fibrosis, and itch-associated conditions[17-21]. IL-31 specifically binds to a heterodimeric receptor complex formed by the IL-31 receptor alpha (IL-31RA) and oncostatin-M specific receptor beta (OSMRβ) and signals through mitogen-activated protein kinase (MAPK) and Janus kinase (JAK/STAT)[22,23]. IL-31 is predominantly produced by hematopoietic cells, whereas its receptor, IL-31RA, is predominantly expressed in non-hematopoietic epithelial and smooth muscle cells (SMC)[24-27]. Although studies have reported an association between the expression of both IL-31 and IL-31RA with severe asthma[28,29], the focus of most prior studies has been on tissue remodeling observed in diseases such as pulmonary fibrosis. For example, we recently published that the absence of IL-31RA had a protective effect against the impairment of lung function in the bleomycin mouse model of pulmonary fibrosis[30]. Moreover, studies from our group and others have shown that Th2 cytokines, IL-4 and IL-13, augment the expression of IL-31RA in immune cells, including macrophages[14,31]. However, the role of the IL-31-IL-31RA axis in the induction of AHR or airway inflammation during allergic asthma remains unexplored.

Here, we provide evidence that both Th1 and Th2 cytokines (IFNγ, IL-4, and IL-13) upregulate IL-31RA but not IL-31 in SMC and lungs during allergic asthma. This intriguing observation led us to further investigate the impact of IL-31RA deficiency on Th1/Th2 cytokine-driven SMC contractility and AHR using cell cultures and animal models in vivo. We identified a mechanism in which IL-31RA functions as a positive regulator of muscarinic acetylcholine receptor 3 (CHRM3) expression, calcium elevation, and MLC phosphorylation to augment agonist-induced contractility of SMC and AHR. Considering a predominant role in regulating airway contractility, CHRM3 is a well-known therapeutic target to relieve bronchoconstriction and AHR in obstructive airway diseases such as asthma and COPD. Thus, the current findings may facilitate the design of therapies with greater efficacy to target AHR downstream of both Th1/Th2 cytokines in asthma.

## Results
### Upregulation of IL-31RA promotes AHR in allergic asthma
Loss of type II IL-4 receptor signaling attenuates the pathological features of asthma and expression of IL-31RA during *Schistosoma mansoni* soluble egg antigen (SEA)-induced allergic asthma[13,31-34]. We hypothesized that in a house dust mite (HDM)-induced model of allergic asthma, the loss of IL-31RA would block the development of some, if not all, pathological features of allergic asthma. To elucidate the role of IL-31RA in allergic asthma, IL-31RA knockout and wild-type mice were sensitized twice with intraperitoneal (i.p.) HDM followed by intratracheal (i.t.) challenge with HDM a week later for two consecutive days (Fig. 1a). The control mice were sensitized and challenged with saline. As expected, the expression of IL-31RA was not detected in IL-31RA knockout (*IL-31RA*⁻/⁻) mice (Fig. 1b). Sensitization and challenge with HDM or saline did not modify the gene expression of *IL-31* in either wild-type or *IL-31RA*⁻/⁻ mice (Fig. 1c), but significantly elevated *IL-31RA* in wild-type mice compared to saline-treated mice (Fig. 1b). To identify whether IL-31RA-expressing cells accumulate in the airways of HDM-challenged mice, the lung sections from wild-type mice treated with saline or HDM were immunostained for IL-31RA. We found increased staining for IL-31RA in the cells localized in the peri-bronchial smooth muscle cell regions with limited or no staining in airway epithelial cells in HDM-challenged mice compared to saline-treated mice (Fig. 1d). To

determine whether increased IL-31RA expression detected in the mouse model of asthma is a feature of human asthma we assessed IL-31RA lung sections from asthmatics and healthy controls (*n* = 8/group). Notably, we found IL-31RA staining was elevated in cells that appear like smooth muscle cells with spindle-shaped nuclei in peri-bronchial and peri-vascular areas of asthmatic human lungs compared to healthy controls (Fig. 1e). We observed no immunostaining with isotype IgG antigen in the lung sections of human or mouse (Supplementary Fig. 1).

IL-31RA is predominantly expressed in ASMC that are critical in airway contractility and AHR. Therefore, we further assessed the role of IL-31RA in regulating AHR in mice. The loss of IL-31RA was sufficient to attenuate dose-dependent methacholine (MCh)-induced AHR in *IL-31RA*⁻/⁻ mice compared to wild-type mice sensitized and challenged with HDM (Fig. 1f). Compared to wild-type mice, *IL-31RA*⁻/⁻ mice treated with saline showed a significant reduction in lung resistance with increasing doses of MCh. To assess whether the observed decrease in AHR was due to impairment of airway contraction, we prepared precision-cut lung slices (PCLS) from wild-type and IL-31RA knockout mice and tested dose-dependent MCh-induced airway contraction in real-time. MCh-induced airway contraction was decreased in PCLS prepared from the lungs of *IL-31RA*⁻/⁻ mice compared to that in wild-type mice (Fig. 1g and Supplementary Fig. 2). Also, we performed AHR measurements using whole-body plethysmography to assess differences in lung function at baseline in naive wild-type mice compared to *IL-31RA*⁻/⁻ mice. Loss of IL-31RA was sufficient to attenuate AHR, corroborating AHR data obtained using flexiVent and the PCLS methods (Fig. 1f, g and Supplementary Fig. 3). Furthermore, we used a collagen gel contraction assay to measure carbachol-induced contraction of ASMC isolated from the trachea of wild-type and *IL-31RA*⁻/⁻ mice. Collagen gels embedded with wild-type ASMC showed higher contraction compared to *IL-31RA*⁻/⁻ ASMC within 30 minutes in floating collagen gels (Fig. 1h). These studies collectively demonstrate that IL-31RA is upregulated in ASMC (murine and human) in asthma, and IL-31RA regulates MCh-induced ASMC contractility, bronchonstriction, and AHR.

### Inflammation and goblet cell hyperplasia are IL-31RA independent
Mice primed and challenged with HDM develop strong Th2 responses, peribronchial inflammation, and goblet cell hyperplasia[35,36]. To determine whether IL-31RA was critical for the development of airway inflammation, we stained the lung sections of wild-type and IL-31RA⁻/⁻ mice challenged with HDM or saline with hematoxylin and eosin (H&E) (Fig. 2a). As expected, the HDM challenge induced robust airway inflammation with immune cell infiltration in wild-type mice, as confirmed by quantitative measurement of the total bronchoalveolar lavage (BAL) cells (Fig. 2b, c). Notably, the *IL-31RA*⁻/⁻ mice that were challenged with HDM developed airway inflammation similar to wild-type mice, indicating no effect of IL-31RA deficiency on allergic inflammatory response in airways. We further investigated whether the loss of IL-31RA had any effect on the infiltration of specific immune cell types into the airspaces by differential counting of the immune cell types in BAL. Consistent with the above findings, we observed no significant differences in the number of eosinophils, macrophages, lymphocytes, and neutrophils between wild-type and *IL-31RA*⁻/⁻ mice challenged with either saline or HDM (Fig. 2c).

Accumulation of mucus-secreting goblet cells or goblet cell hyperplasia is another important hallmark of allergic asthma observed in both human and animal models of allergic asthma[37-39]. To determine whether there were differences in goblet cell hyperplasia, we performed Alcian blue periodic acid-Schiff (ABPAS) staining to detect mucus-producing goblet cells in lung sections. Both wild-type and *IL-31RA*⁻/⁻ mice displayed increased goblet cell hyperplasia when challenged with HDM compared to saline-challenged animals (Figs. 2d, e). These findings suggest that IL-31RA contributes to the development of

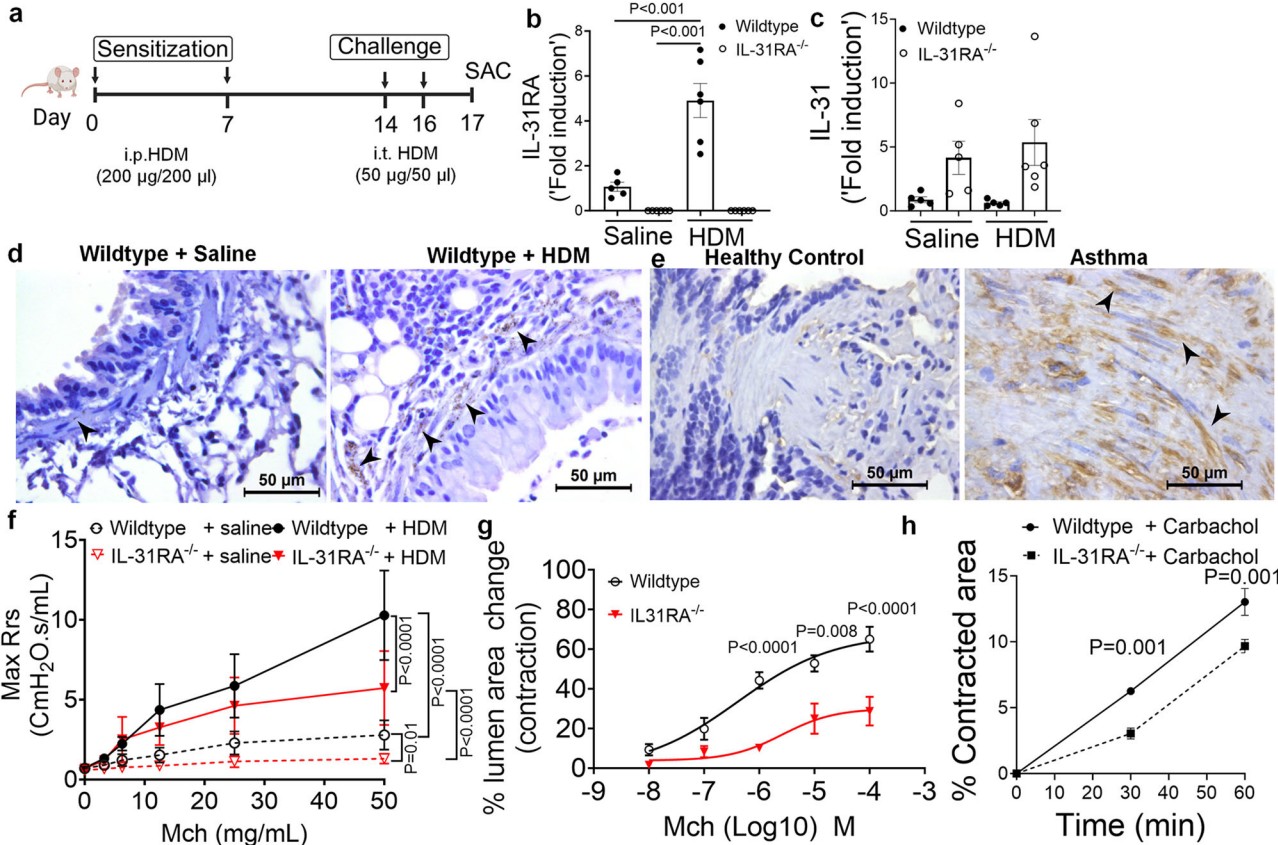

**Fig. 1 | Loss of IL-31RA attenuates house dust mite (HDM)-induced airway hyperresponsiveness. a** Schemata of HDM-induced allergic asthma model. Image was created with biorender.com. **b** Quantification of IL-31RA transcripts in the lungs of wildtype ($n = 5$) and IL-31RA$^{-/-}$ ($n = 5$) mice treated with saline or wildtype ($n = 6$) and IL-31RA$^{-/-}$ ($n = 6$) mice treated with HDM. Data shown as mean ± SEM. Two-way ANOVA test was used. **c** Quantification of IL-31 transcripts in the lungs of wildtype ($n = 5$) and IL-31RA ($n = 5$) mice treated with saline or wildtype ($n = 5$) and IL-31RA ($n = 6$) mice treated with HDM. Data shown as mean ± SEM. Two-way ANOVA test was used. **d** Representative images of immunohistochemical staining for IL-31RA on lung sections from wild-type mice treated with saline ($n = 4$) or HDM ($n = 4$). Scale bar, 50 μm. **e** Representative images of immunohistochemical staining for IL-31RA on lung sections from healthy controls ($n = 8$) and asthmatics ($n = 8$). Arrow heads highlight the IL-31RA staining (brown) in smooth muscle cells with spindle-shaped nuclei. Scale bar, 50 μm. **f** Measurement of resistance with increasing doses of methacholine (MCh) in wildtype ($n = 7$) and IL-31RA$^{-/-}$ ($n = 8$) mice treated with saline or wildtype ($n = 5$) and IL-31RA$^{-/-}$ ($n = 8$) mice treated with HDM using Flexi-Vent. Data are shown as mean ± SEM. A two-way ANOVA test was used. **g** The percent of contraction of airways with increasing doses of MCh compared to the baseline area of airways between wild-type ($n = 8$) and IL-31RA$^{-/-}$ ($n = 6$) mice. A two-way ANOVA test was used. Data shown as mean ± SEM. **h** The percent contraction of collagen gels embedded with airway smooth muscle cells from wild-type ($n = 3$) and IL-31RA$^{-/-}$ ($n = 3$) mice in culture media and treated with carbachol (10 μM) for 60 min. Data shown as mean ± SEM. Unpaired $t$ test was used. At least two independent experiments produced similar results. Source data are provided as a Source Data file.

AHR but has no significant effect on airway inflammation and mucus production in HDM-induced allergic asthma.

IL-31RA functions as a negative regulator of Th2 responses in the lungs[40]. However, the role of IL-31RA in the development of Th2 responses during allergic asthma remains unclear. To determine whether there are alterations in the expression of asthma-associated genes in the absence of IL-31RA, we measured the transcript levels of Th2 cytokines, inflammatory cytokines, Th2 responses, and goblet cell hyperplasia-associated genes. We observed a significant increase in Th2 cytokine gene expression, including *IL-4, IL-5,* and *IL-13*, in wild-type mice challenged with HDM compared to that with saline (Fig. 3a). However, HDM-induced increase in Th2 cytokine gene expression was similar in the lungs of wild-type and *IL-31RA*$^{-/-}$ mice, suggesting that the absence of IL-31RA had no effect on HDM-induced Th2 cytokine expression. Similarly, the loss of IL-31RA had no effect on the expression of chemokine and cytokine genes *(CCL11, CCL24, and IL-10)* and Th2 response-associated genes, including *ARG1, CHI3L3,* and *FIZZ1* (Fig. 3b, c). Consistent with ABPAS staining, we observed no effect of IL-31RA deficiency on HDM-induced expression of *GOB5* and *MUC5AC* (Fig. 3d). *IL-6, oncostatin M (OSM),* and *IL-31* are members of the IL-6

family of cytokines known to regulate AHR in allergic asthma[41–43]. Thus, we explored the levels of *OSM* and *IL-6* in the lungs of wild-type and IL-31RA$^{-/-}$ mice (Supplementary Fig. 4). We observed no significant differences in the expression of *IL-6* and *OSM* between wild-type and *IL-31RA*$^{-/-}$ mice treated with either saline or HDM, suggesting that the absence of IL-31RA did not affect the expression of *OSM* and *IL-6*. To validate the above transcriptional changes, we measured the protein levels of IL-4, IL-13, IL-5, IL-6, IFNγ, TNF-α and IL-10 cytokine levels in BAL fluid of wild-type and IL-31RA$^{-/-}$ mice sensitized and challenged with HDM or saline. The levels of IL-4, IL-13, IL-5, and IL-6 were significantly elevated in BAL fluid of HDM-challenged wild-type and *IL-31RA*$^{-/-}$ mice compared to saline-treated mice (Fig. 3e). Similar to gene transcript levels in the lungs, we observed no significant differences in Th2 cytokine protein levels in wild-type mice compared to *IL-31RA*$^{-/-}$ mice treated with HDM (Fig. 3e).

## IL-31RA is essential to induce AHR during SEA-induced allergic asthma

To further establish the role of IL-31RA in allergic asthma, we used an alternative mouse model of SEA-induced allergic asthma. Similar to

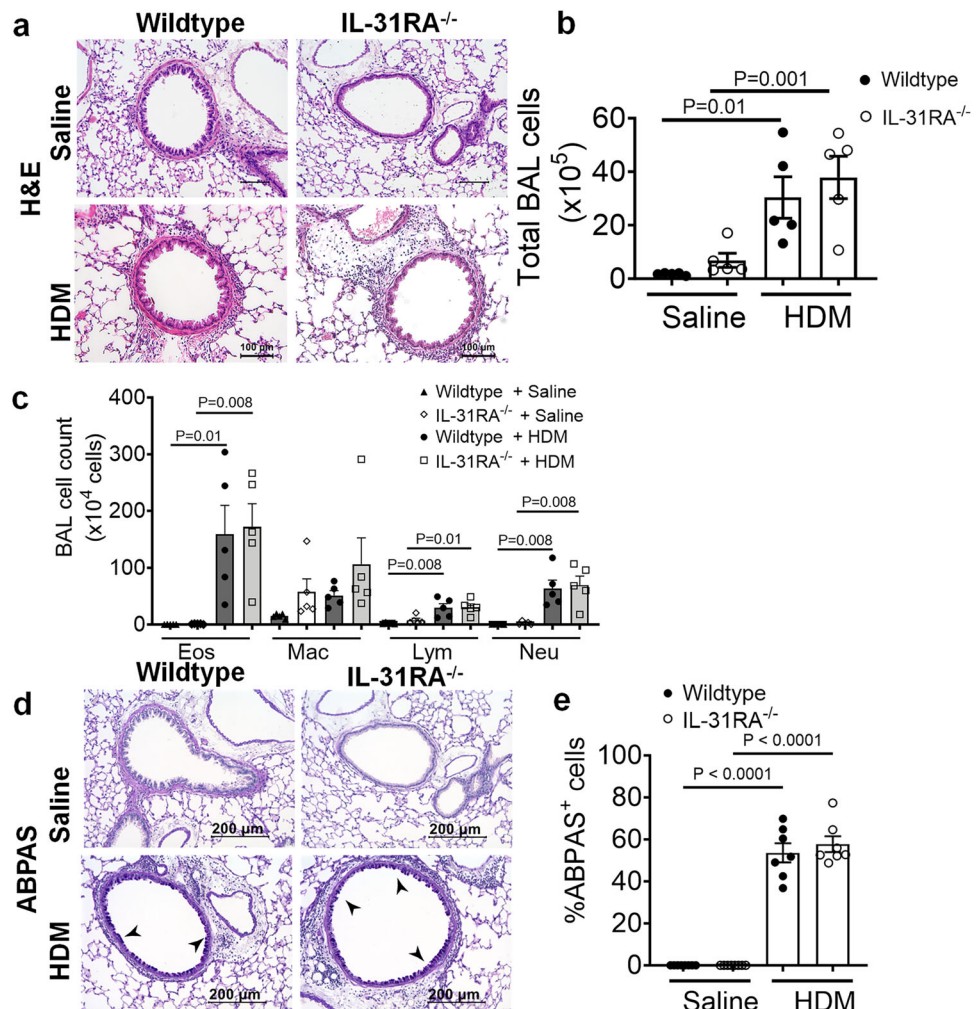

**Fig. 2 | Loss of IL31RA has no effect on house dust mite (HDM)-induced airway inflammation and goblet cell hyperplasia. a** Representative image of hematoxylin and eosin-stained lung sections from wildtype ($n = 5$) and IL-31RA$^{-/-}$ ($n = 5$) mice treated with saline or HDM. Images were taken at ×20 magnification, scale bar 100 μm. **b**, **c** Total bronchoalveolar lavage (BAL) cell number and the differential cell count of BAL cells of wild-type and IL-31RA$^{-/-}$ mice treated with saline ($n = 5$) or HDM ($n = 5$). Eos, eosinophils; Mac, macrophages; Lym, lymphocytes; and Neu, neutrophils. One-way ANOVA was used. **d** Representative images of Alcian blue periodic acid-Schiff (ABPAS) staining of lung sections from wildtype ($n = 7$) and IL-31RA$^{-/-}$ ($n = 7$) mice treated with saline or HDM. Images were taken at ×20 magnification, scale bar 100 μm. **e** The percent of ABPAS-positive cells normalized to total cell in the airways of wild-type and IL-31RA$^{-/-}$ mice treated with saline ($n = 7$) or HDM ($n = 7$). Data are shown as mean ± SEM. One-way ANOVA was used. At least two independent experiments produced similar results. Source data are provided as a Source Data file.

HDM, extract of *Schistosoma mansoni* is a potent inducer of AHR, airway inflammation, and Th2 immune responses even without the use of an adjuvant[21,44,45]. As shown in the schematic, wild-type and *IL-31RA*$^{-/-}$ mice were sensitized twice and consecutively challenged with SEA (Fig. 4a). The absence of IL-31RA gene expression was confirmed by measuring transcripts in the lungs of mice challenged with SEA or saline, and no IL-31RA expression was detected in IL31RA$^{-/-}$ mice (Fig. 4b). Consistent with the findings using the HDM model, sensitization and challenge with SEA resulted in a significant increase in the gene expression of *IL-31RA* but not *IL-31* in the wild-type mice (Fig. 4b, c).

To assess whether the loss of IL-31RA attenuates SEA-induced AHR, we measured AHR in both wild-type and *IL-31RA*$^{-/-}$ mice sensitized and challenged with SEA or saline (Fig. 4d, e). Loss of IL-31RA attenuated airway responsiveness in mice treated with saline (Fig. 4d). Similar to HDM model, we observed a significant decrease in AHR in *IL-31RA*$^{-/-}$ mice compared to wild-type mice sensitized and challenged with SEA (Fig. 4e). To evaluate the changes in airway inflammation, lung sections of wild-type and *IL31RA*$^{-/-}$ mice challenged with SEA or saline were stained with H&E. As observed in the HDM model, SEA

challenge induced robust airway inflammation with immune cell infiltration in both wild-type and *IL-31RA*$^{-/-}$ mice (Fig. 4f). Similarly, the airways of *IL-31RA*$^{-/-}$ mice that were challenged with SEA showed an accumulation of mucus-producing goblet cells similar to wild-type mice, as seen in ABPAS-stained lung sections (Fig. 4g).

To determine whether IL-31RA-induced AHR is regulated by asthma-associated gene networks, we measured the expression of Th1 and Th2 cytokines. The expression of *IL-4* and *IL-5* were elevated in both wild-type and *IL-31RA*$^{-/-}$ mice following sensitization and challenge with SEA (Supplementary Fig. 5a). Similarly, the transcripts of genes associated with inflammation (*CCL11, CCL24, and IL-10*) and Th2 responses (*ARG1, CHI3L3, and FIZZ1*) remained elevated in *IL-31RA*$^{-/-}$ mice, and no differences were observed when compared to wild-type mice treated with SEA (Supplementary Fig. 5b, c). The gene transcripts associated with goblet cell hyperplasia were elevated in both wild-type and *IL-31RA*$^{-/-}$ mice treated with SEA (Supplementary Fig. 5d). In summary, in the two alternative mouse models of allergic asthma, the absence of IL-31RA altered AHR but did not appear to affect airway inflammation, Th2 responses, and goblet cell hyperplasia.

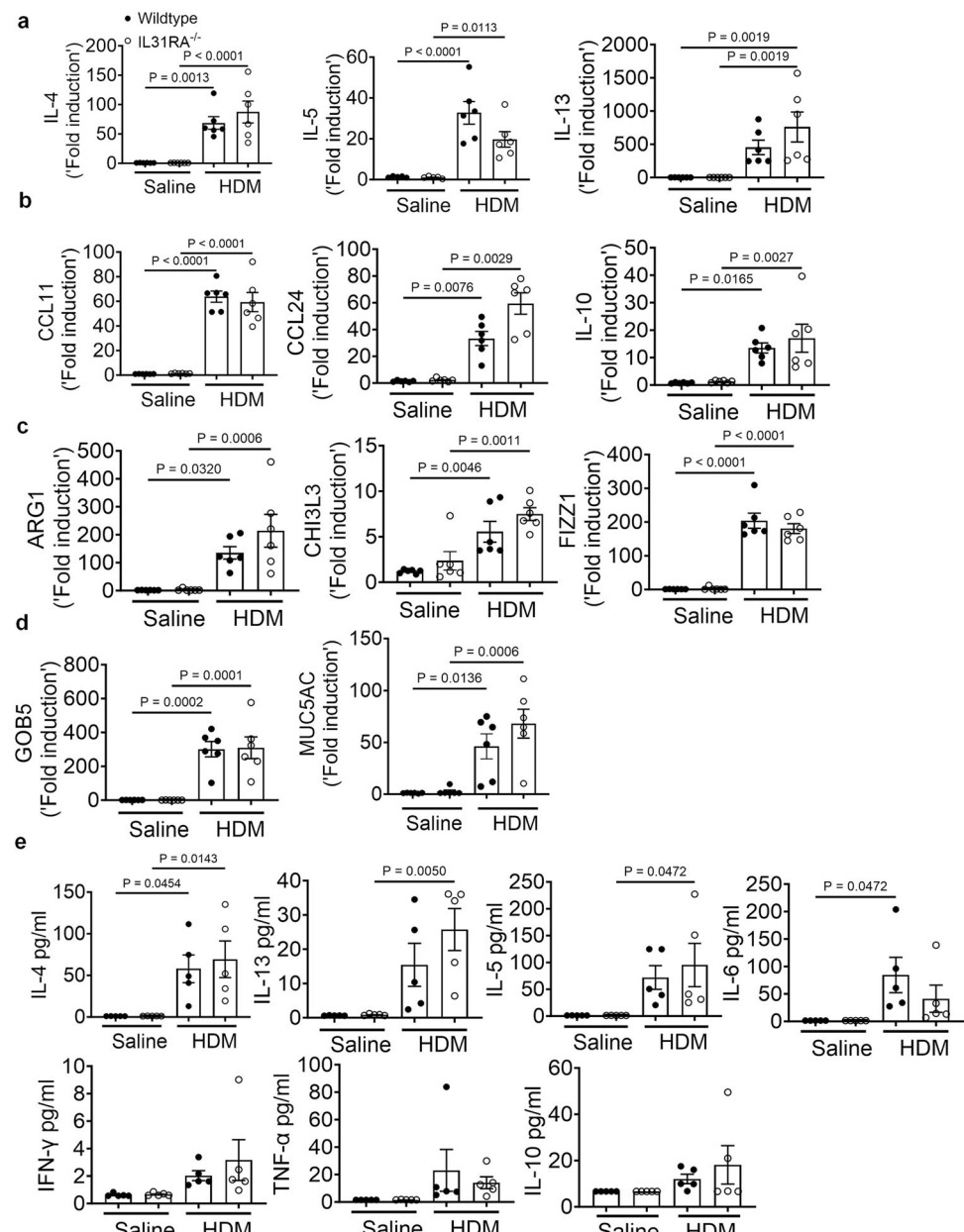

**Fig. 3 | Loss of IL-31RA has no effect on Th2 responses and goblet cell hyperplasia during house dust mite (HDM)-induced allergic asthma. a** Quantification of Th2 cytokine transcripts including IL-4, IL5, and IL-13 in the whole lung tissue of wild-type and IL-31RA$^{-/-}$ mice treated with saline ($n = 6$) or HDM ($n = 6$) using RT-PCR. Data are shown as mean $\pm$ SEM. One-way ANOVA was used. **b** Quantification of chemokines and inflammatory cytokines including CCL11, CCL24, and IL-10 in the whole lungs of wild-type and IL-31RA$^{-/-}$ mice treated with saline ($n = 6$) or HDM ($n = 6$) using RT-PCR. Data are shown as mean $\pm$ SEM. One-way ANOVA was used. **c** Quantification of Th2 response-associated gene transcripts including CHI3L3, ARG1, and FIZZ1 in the total lungs of wild-type and IL-31RA$^{-/-}$ mice treated with saline ($n = 6$) or HDM ($n = 6$) using RT-PCR. Data are shown as mean $\pm$ SEM. One-way ANOVA was used. **d** Quantification of GOB5 and MUC5AC gene transcripts in the whole lungs of wild-type and IL-31RA$^{-/-}$ mice treated with saline ($n = 6$) or HDM ($n = 6$) using RT-PCR. Data are shown as mean $\pm$ SEM. One-way ANOVA was used. **e** Concentration of cytokines IL-4, IL-13, IL-5, IL-6, IFNγ, TNF-α and IL-10 in the BAL fluids of wild-type and IL-31RA$^{-/-}$ mice treated with saline ($n = 6$) or HDM ($n = 6$) using the cytometric bead assay LEGENDplex. Data are shown as mean $\pm$ SEM. One-way ANOVA was used. At least two independent experiments produced similar results. Source data are provided as a Source Data file.

## IL-31 is dispensable to induce AHR, inflammation, and Th2 responses

IL-31RA is a cognate binding receptor of IL-31 that recruits OSMRβ to form a high-affinity binding receptor complex and signals through the JAK/STAT pathway[22,23]. Previous studies have demonstrated that increased IL-31 signaling through IL-31RA can result in uncontrolled inflammation and tissue remodeling in multiple tissues, including the lungs and skin[18,19]. However, we observed no significant increase in IL-31 levels in the lungs during HDM- and SEA-induced allergic asthma. To assess the potential pathological effects of IL-31 in asthma, the lungs of

wild-type mice were intratracheally treated with saline or IL-31, and changes in AHR and inflammation were assessed (Fig. 5a). Notably, we observed no significant changes in AHR between saline and IL-31 treated wild-type mice (Fig. 5b). Similarly, we observed no significant changes in tissue inflammation as assessed by H&E-stained lung sections (Fig. 5c). To confirm the signaling effects of IL-31, we measured the expression of the IL-31-driven gene suppressor of cytokine signaling 3 (*SOCS3*) in the lungs of mice treated with saline and IL-31. Previously published studies have shown that IL-31 upregulates *SOCS3*;[46] similarly, the expression of *SOCS3* was significantly

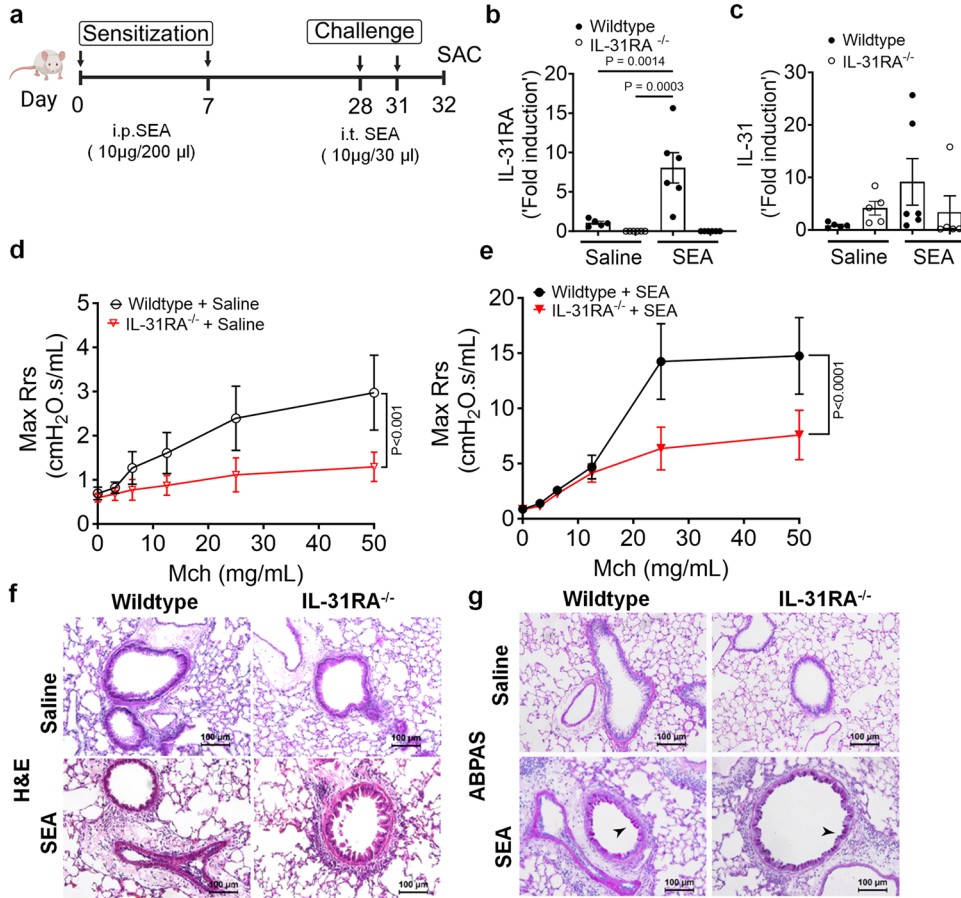

**Fig. 4 | Loss of IL31RA attenuates AHR but not inflammation and goblet cell hyperplasia during SEA-induced allergic asthma. a** Schemata of SEA-induced allergic asthma model. Image was created with biorender.com. **b** Quantification of IL-31RA transcripts in the lungs of wildtype ($n = 5$) and IL-31RA$^{-/-}$ ($n = 6$) mice treated with saline or wild-type ($n = 6$) and IL-31RA$^{-/-}$ ($n = 6$) mice treated with SEA. Data shown as mean ± SEM. Two-way ANOVA test was used. **c** Quantification of IL-31 transcripts in the lungs of wildtype ($n = 5$) and IL-31RA$^{-/-}$ ($n = 5$) mice treated with saline or wild-type ($n = 6$) and IL-31RA$^{-/-}$ ($n = 5$) mice treated with SEA. Data shown as mean ± SEM. Two-way ANOVA test was used. **d**, **e** Measurement of resistance with increasing doses of methacholine (MCh) in wildtype ($n = 6$) and IL-31RA$^{-/-}$ ($n = 7$) mice treated with saline or wildtype ($n = 5$) and IL-31RA$^{-/-}$ ($n = 4$) mice treated with SEA using FlexiVent. Data are shown as mean ± SEM. A two-way ANOVA test was used. **f** Representative images of hematoxylin and eosin-stained lung sections from wildtype ($n = 5$) and IL-31RA$^{-/-}$ ($n = 5$) mice treated with saline or SEA. Images were taken at ×20 magnification. Scale bar, 100 μm. **g** Representative images of Alcian blue periodic acid-Schiff staining of lung sections from wildtype ($n = 5$) and IL-31RA$^{-/-}$ ($n = 5$) mice treated with saline or SEA. Images were taken at ×20 magnification. Scale bar, 100 μm. At least two independent experiments produced similar results. Source data are provided as a Source Data file.

upregulated in lungs of IL-31-treated mice compared to the saline-treated mice (Fig. 5d). To determine the effects of IL-31 on the expression of asthma-associated genes, we quantified the expression of genes associated with inflammation (*IFNγ, TNF-α, IL-6, and IL-17*) and Th2 responses (*IL-4, IL-13, ARG1, MUC4, and MUCSAC*). Notably, we observed no significant changes in the expression of genes associated with either inflammation or Th2 responses (Figs. 5e, f).

Furthermore, we evaluated airway contractility induced by IL-31 using PCLS. As demonstrated in previous studies, we observed a significant increase in the contractility of airways of wild-type mice treated with IL-13 compared to saline-treated mice in a dose-dependent manner with MCh (Supplementary Fig. 6). Conversely, we observed no significant effect of IL-31 on the MCh-induced contractility of wild-type airways (Fig. 5g). Furthermore, we observed no significant changes in the kinetics of contraction of collagen gels embedded with ASMC, which were treated with either media alone or IL-31. However, we observed a significant increase in the contraction of collagen gels embedded with ASMC and treated with IL-13 compared to that with media alone (Fig. 5h). Thus, in contrast to IL-31RA, the results suggest that IL-31 is not involved in modifying AHR, and the reduced AHR observed in the absence of IL-31RA could be due to other alternative mechanisms that need to be identified.

## Th2 cytokines upregulate IL-31RA to induce AHR

Previous studies from our lab and others have shown that both IL-4 and IL-13 induce IL-31RA expression in macrophages and lung tissues via the type II IL-4 receptor and STAT6 signaling[31]. To determine whether IL-4 or IL-13 can induce IL-31RA expression in ASMC, we treated ASMC with increasing doses of IL-4 or IL-13 and quantified the expression of *IL31RA*. Consistent with our earlier findings in macrophages, the expression of *IL-31RA* significantly increased in ASMC treated with IL-4 or IL-13 compared to that with media (Fig. 6a, b). IL-13 is a potent inducer of AHR, inflammation, and goblet cell hyperplasia in mice[11]. To determine whether IL-13-induced AHR is IL-31RA dependent, we treated both wild-type and *IL-31RA*$^{-/-}$ mice with IL-13 and assessed AHR and other pathological and molecular changes that are relevant to asthma. Notably, the loss of IL-31RA was sufficient to attenuate IL-13-induced AHR compared to wild-type mice exposed to increasing doses of MCh (Fig. 6c). Similarly, the contraction of collagen gels embedded with ASMC from *IL-31RA*$^{-/-}$ mice significantly reduced compared to wild-type mice with IL-13 treatment (Fig. 6d). In contrast, H&E staining of lung sections suggested that the loss of IL-31RA had no effect on IL-13-driven airway inflammation in wild-type and *IL31RA*$^{-/-}$ mice (Fig. 6e). Quantitative assessment of inflammatory chemokines and cytokines suggested that the loss of IL-31RA had no effect on the expression of

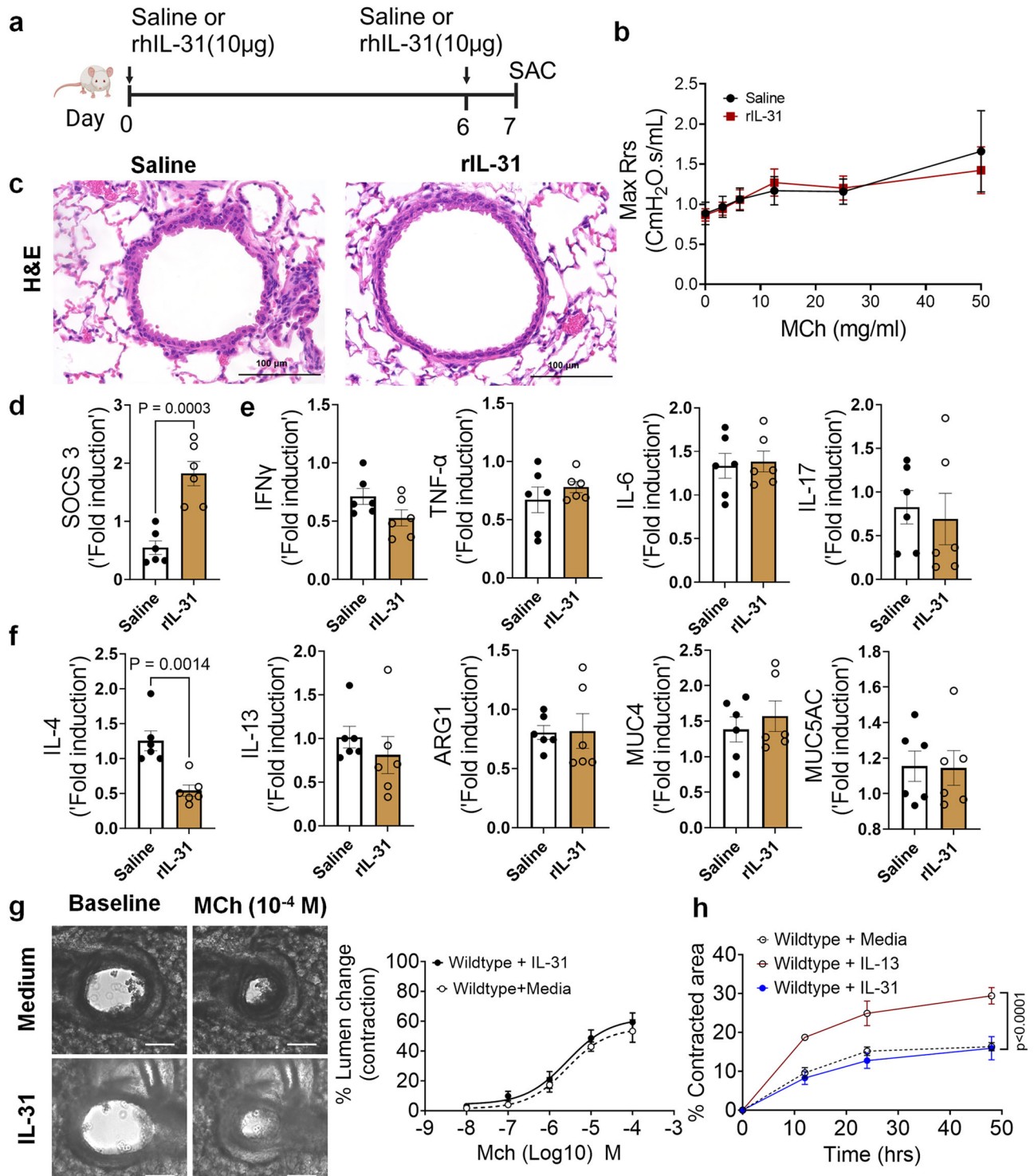

CCL11, CCL24, and IL-17 (Fig. 6f). Similarly, we observed no defects in the expression of Th2 cytokines (IL-4 and IL-5) or genes associated with Th2 responses *(ARG1, CHI3L3, and FIZZ1)* in *IL-31RA*[−/−] mice compared to wild-type mice treated with IL-13 (Figs. 6g, h).

To evaluate whether IL-13-induced goblet cell hyperplasia was altered in the absence of IL-31RA, we assessed the accumulation of goblet cells in the airways and quantified the transcripts associated with goblet cell hyperplasia. As shown in Fig. 6i, goblet cell accumulation was similar between wild-type and *IL31RA*[−/−] mice treated with IL-13. In support of this, the expression of *GOB5* and *MUC5AC* remained similar between IL-13 treated wild-type and *IL31RA*[−/−] mice (Fig. 6j).

Thus, IL-31RA expression is essential to mediate IL-13-induced AHR but dispensable for inflammation and goblet cell hyperplasia.

## IFNγ upregulates IL-31RA to induce AHR

IFNγ is a key Th1 cytokine implicated in elevated AHR in patients with severe asthma and those with mixed Th1/Th2 cytokine phenotypes[47,48]. To determine whether IFNγ induces the expression of IL-31RA in ASMC, we isolated ASMC from the tracheas of wild-type mice and treated them with increasing concentrations of IFNγ. The expression of IL-31RA is significantly increased by IFNγ in a dose dependent manner (Fig.7a). However, we observed no effect of IL-31 on the expression of

**Fig. 5 | IL-31 is dispensable for the induction of AHR, inflammation, and Th2 responses. a** Schemata showing intratracheal administration of IL-31 or saline in wild-type mice. Image was created with biorender.com. **b** Measurement of resistance with increasing doses of methacholine (MCh) in wild-type mice treated with saline ($n = 5$) or IL-31 ($n = 5$) using FlexiVent. Data are shown as mean ± SEM. Two-way ANOVA was used. The data is representative of two independent experiments with no statistical significance between groups. **c** Representative images of hematoxylin and eosin-stained lung sections from wildtype mice treated with IL-31 ($n = 5$) or saline ($n = 5$). Images were captured at ×20 magnification. Scale bar, 100 μm. **d** Quantification of IL-31-induced SOCS3 gene expression in the whole lung tissue of wild-type mice treated with saline ($n = 6$) or IL-31 ($n = 6$). Unpaired $t$ test was used. Data shown as mean ± SEM. **e, f** Quantification of inflammation-associated gene transcripts *IFNγ, TNFα, IL-6*, and *IL-17* and Th2-associated gene transcripts including *IL-4, IL-13, ARG1, MUC4*, and *MUC5AC* in whole lungs of wild-type mice treated with

saline ($n = 6$) or IL-31 ($n = 6$). Unpaired $t$ test was used and no significance found between groups. Data shown as mean ± SEM. **g** Representative images of precision cut lung sections (PCLS) from wildtype mice treated with media or IL-31 (500 ng/ mL) for 24 h. Airway contractility was measured in response to MCh ($10^{-4}$ M) compared to baseline lumen area. The percent of airway lumen area contraction with increasing doses of MCh was calculated for media ($n = 5$) and IL-31-treated ($n = 8$) PCLS from wildtype mice. Two-way ANOVA test was used. Data shown as mean ± SEM. **h** The percent contraction of collagen gels embedded with airway smooth muscle cells from wild-type mice that were treated with media, IL-13 (50 ng/mL) or IL-31 (500 ng/mL). The percent contraction was measured at different time points compared to baseline. Two-way ANOVA was used. $n = 4$/group. Data shown as mean ± SEM. At least two independent experiments produced similar results. Source data are provided as a Source Data file.

IL-31RA in human ASMC (Fig. 7b). To determine the effects of IFNγ on asthma phenotypes, we treated wild-type mice with saline or IFNγ and measured the changes in AHR, inflammation, and goblet cell hyperplasia. We observed a significant increase in AHR in wild-type mice treated with IFNγ compared to that with saline (Fig. 7c). In addition, we observed a significant increase in peri-bronchial inflammation in wild-type mice treated with IFNγ compared to that with saline (Fig. 7d). To establish the quantitative gene expression changes induced by IFNγ, we measured *IRF1, IRF7*, and *STAT1* and observed a significant increase in the expression of IFNγ-specific genes in the lungs (Supplementary Fig. 7).

To determine the effects of IFNγ on asthma phenotypes, we measured the expression of genes associated with inflammation and Th2 responses. In support of the substantial inflammation observed after IFNγ treatment, we observed a significant increase in *CCL11* and *IL-17* expression (Fig. 7e). As anticipated, IFNγ treatment resulted in negative regulation of Th2 cytokines and Th2 response-associated gene expression, including *IL-4, IL-5, CCL24, ARG1, and CHI3L3* (Fig. 7f, g). Similar to IL-13, IFNγ was able to induce the accumulation of goblet cells in the airways with increased expression of *GOB5* and *MUC5AC* (Figs. 7h, i). Despite the reduced expression of several Th2-associated asthma genes, IFNγ treatment induced AHR, inflammation, and goblet cell hyperplasia.

To determine whether the expression of IL-31RA was critical for IFNγ-induced AHR, inflammation, and goblet cell hyperplasia, *IL31RA⁻/⁻* mice were intratracheally treated with IFNγ. Notably, IFNγ-induced AHR was significantly attenuated in *IL31RA⁻/⁻* mice compared to that in wild-type mice (Fig. 8a and Supplementary Fig. 8). However, as noted with IL-13, increases in peribronchial inflammation and the expression of inflammatory cytokine genes, such as *CCL11, CCL24*, and *IL-17* were similar between wild-type and *IL-31RA⁻/⁻* mice treated with IFNγ (Figs. 8b, c). Furthermore, we observed no significant differences in the expression of Th2 cytokine genes *(IL-4* and *IL-5)* or the expression of genes associated with Th2 responses, including *ARG1* and *CHI3L3* (Figs. 8d, e).

To evaluate the effects of IL-31RA deficiency on IFNγ-induced goblet cell hyperplasia, we performed ABPAS staining of lung sections from wild-type and *IL-31RA⁻/⁻* mice and quantified *GOB5* and *MUC5AC* expression. Analysis of ABPAS-stained lung sections suggested no change in the number of goblet cells that accumulated in the airways of wild-type and *IL31RA⁻/⁻* mice treated with IFNγ (Fig. 8f). This finding was further substantiated by quantitative PCR data, which showed no quantitative differences in the expression of GOB5 and MUC5AC with the loss of IL-31RA (Fig. 8g). Nevertheless, our in vivo studies demonstrated that despite the development of substantial inflammation and mucus hypersecretion, *IL-31RA⁻/⁻* mice showed attenuated AHR in response to either IL-13 or IFNγ. Overall, our findings convincingly demonstrate that in allergic asthma, IL-31RA, which is induced by both Th1 and Th2, is critically required for the development of AHR but not inflammation and mucus secretion.

## IL-31RA augments CHRM3-driven calcium signaling in ASMC

Muscarinic acetylcholine receptors (CHRMs) are predominantly expressed by structural cells such as smooth muscle cells of airways and play a major role in triggering contraction of airways and AHR[49,50]. To evaluate mechanisms by which IL-31RA induces AHR, we measured the transcripts of five major receptor subtypes including *CHRM1, CHRM2, CHRM3, CHRM4 and CHRM5* in the lungs of wild-type and *IL-31RA⁻/⁻* mice. Quantification of the lung transcripts suggest no significant differences in the transcript levels of CHRMs in the lungs of *IL-31RA⁻/⁻* mice compared to wild-type mice (Fig. 9a). CHRM3 is a dominant receptor subtype expressed in ASMC and coupled to the $G_{q/11}$ family of G proteins to induce calcium signaling and the contraction of ASMC[49,51]. Therefore, we evaluated the changes in the transcripts of *CHRM3* by IL-4 and IL-31 in ASMC of wild-type and IL-31RA⁻/⁻ mice. Neither IL-4 nor IL-31 had a significant effect on the expression of *CHRM3* either in the presence or absence of IL-31RA suggesting other post-transcriptional mechanisms might be involved in IL-31RA-driven contraction of ASMC (Figs. 9b, c). To assess whether IL-31RA alters CHRM3 protein expression, we performed western blot analysis of IL-31RA and CHRM3 in the lung lysates of wild-type and *IL-31RA⁻/⁻* mice that were sensitized and challenged with HDM. Western blot analysis of the total lung lysates shows that the loss of IL-31RA attenuated the CHRM3 protein levels in the lungs of *IL-31RA⁻/⁻* mice compared to wild-type mice challenged with HDM (Fig. 9d, e). To further establish the role of IL-31RA in CHRM3 protein expression in ASMC, we measured the protein levels of CHRM3 in ASMC isolated from wild-type and *IL-31RA⁻/⁻* mice. Notably, we observed a significant decrease in the protein levels of CHRM3 in ASMC isolated from *IL-31RA⁻/⁻* mice compared to wild-type mice (Fig. 9f). Similarly, decrease in *IL-31RA* expression was sufficient to attenuate CHRM3 protein levels in the cell lysates of hTERT cells transfected with IL-31RA-specific siRNA compared to control siRNA for 72 h (Fig. 9g). To assess whether IL-31RA alters CHRM3 protein expression on cell surfaces, we performed western blot analysis of IL-31RA and CHRM3 in plasma membrane enriched fraction using cell surface protein biotinylation assay in HEK293 cells that were transfected with either control or overexpression plasmid for IL-31RA. Western blot analysis of total cell lysates shows overexpression of IL-31RA in HEK293T cells transfected with IL-31RA compared to control plasmid (Supplementary Fig. 9). Notably, we observed a significant increase in CHRM3 protein on the cell surface of HEK293 cells overexpressing IL-31RA compared to cells transfected with the control plasmid or the negative control for cell surface biotinylation (Supplementary Fig. 10). To assess the role of IL-31RA on cell surface CHRM3 expression in human ASMC, we performed a western blot analysis for CHRM3 in the plasma membrane protein-enriched fraction using a cell surface biotinylation assay in hTERT cells transiently transfected with scrambled or IL-31RA-specific siRNA for 72 h. We observed a significant decrease in CHRM3 protein levels normalized to the other cell surface protein ITGB1 in hTERT cells transduced with IL-31RA-specific siRNA compared to control siRNA or the negative control

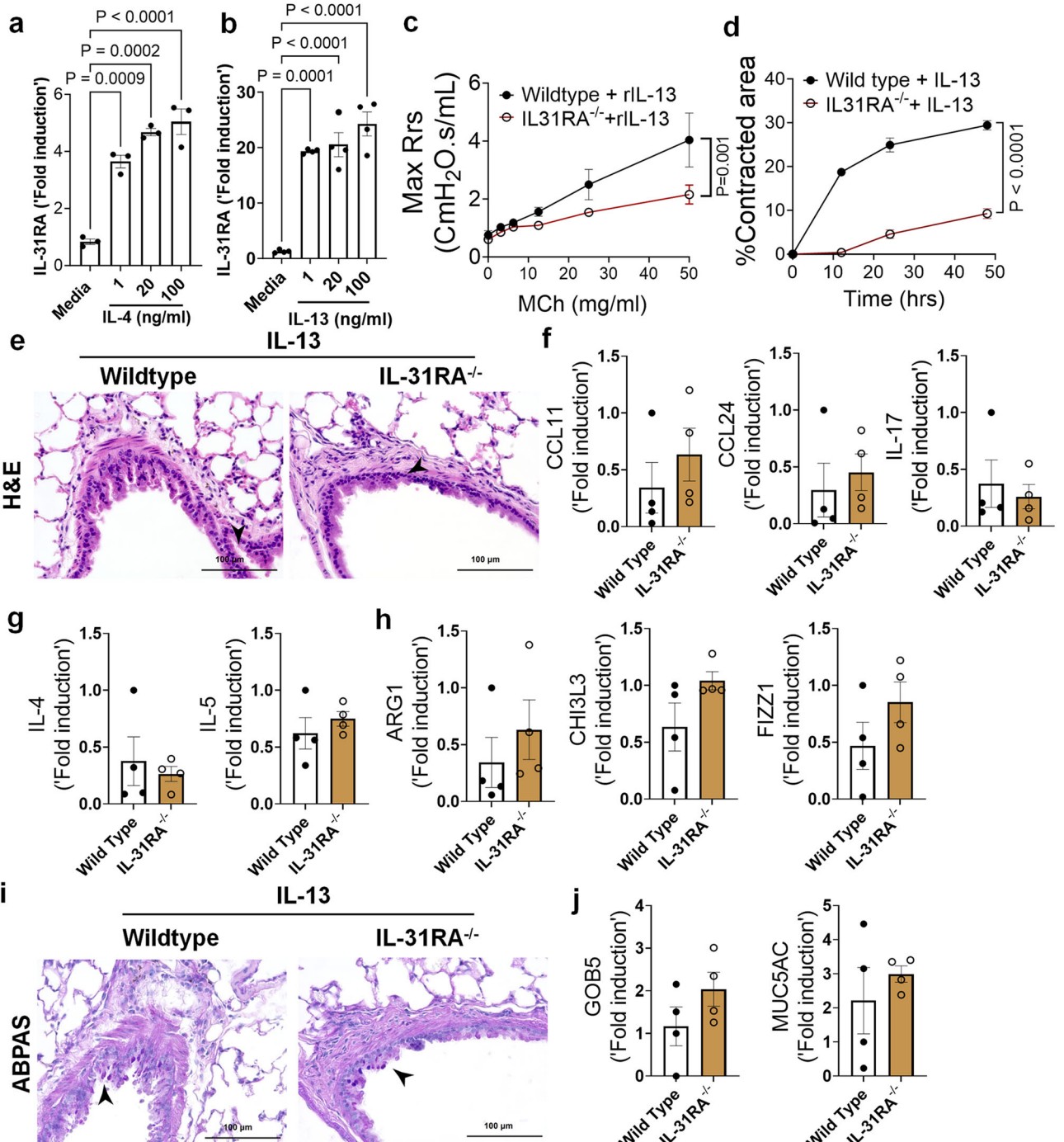

**Fig. 6 | Th2 cytokines upregulate IL-31RA to induce AHR with no effect on inflammation and goblet cell hyperplasia. a, b** Quantification of IL-31RA transcripts in mouse ASMC treated with increasing doses of IL-4 ($n = 3$) and IL-13 ($n = 4$) for 16 h. Two-way ANOVA test was used. Data shown as mean ± SEM. **c** Wildtype ($n = 4$) and IL-31RA$^{-/-}$ ($n = 4$) mice were treated intratracheally with IL-13 on days 0 and 6, and resistance in the lungs was measured with increasing doses of methacholine (MCh) using FlexiVent. Data are shown as mean ± SEM. Two-way ANOVA test was used. **d** ASMC isolated from wildtype ($n = 3$) and IL-31RA$^{-/-}$ ($n = 3$) mice were seeded into collagen gels and treated with IL-13 to measure the contraction of collagen gels after 48 h. Unpaired $t$ test was used. Data shown as mean ± SEM. **e** Representative images of hematoxylin and eosin -stained lung sections from wildtype ($n = 4$) and IL-31RA$^{-/-}$ ($n = 4$) mice treated with IL-13. Images were captured at ×20 magnification. Scale bar, 100 μm. **f, g, h** Quantification of inflammatory

chemokines (*CCL11, CCL24 and IL-17*), Th2 cytokines (*IL-4* and *IL-5*), and Th2 response-associated genes (*ARG1, CHI3L3,* and *FIZZ1*) in the whole lungs of wildtype ($n = 4$) and IL-31RA$^{-/-}$ ($n = 4$) mice treated with IL-13. Data are shown as mean ± SEM. Unpaired $t$ test was used, and no statistical significance observed between groups. **i** Representative images of Alcian blue periodic acid-Schiff-stained lung sections from wildtype ($n = 4$) and IL-31RA$^{-/-}$ ($n = 4$) mice treated with IL-13. Images were captured at ×20 magnification. Scale bar, 100 μm. **j** Quantification of goblet cell hyperplasia associated genes including *GOB5* and *MUC5AC* transcript levels in the total lungs of wildtype ($n = 4$) and IL-31RA$^{-/-}$ ($n = 4$) mice treated with IL-13. Data are shown as means ± SEM. Unpaired $t$ test was used, and no statistical significance observed between groups. At least two independent experiments produced similar results. Source data are provided as a Source Data file.

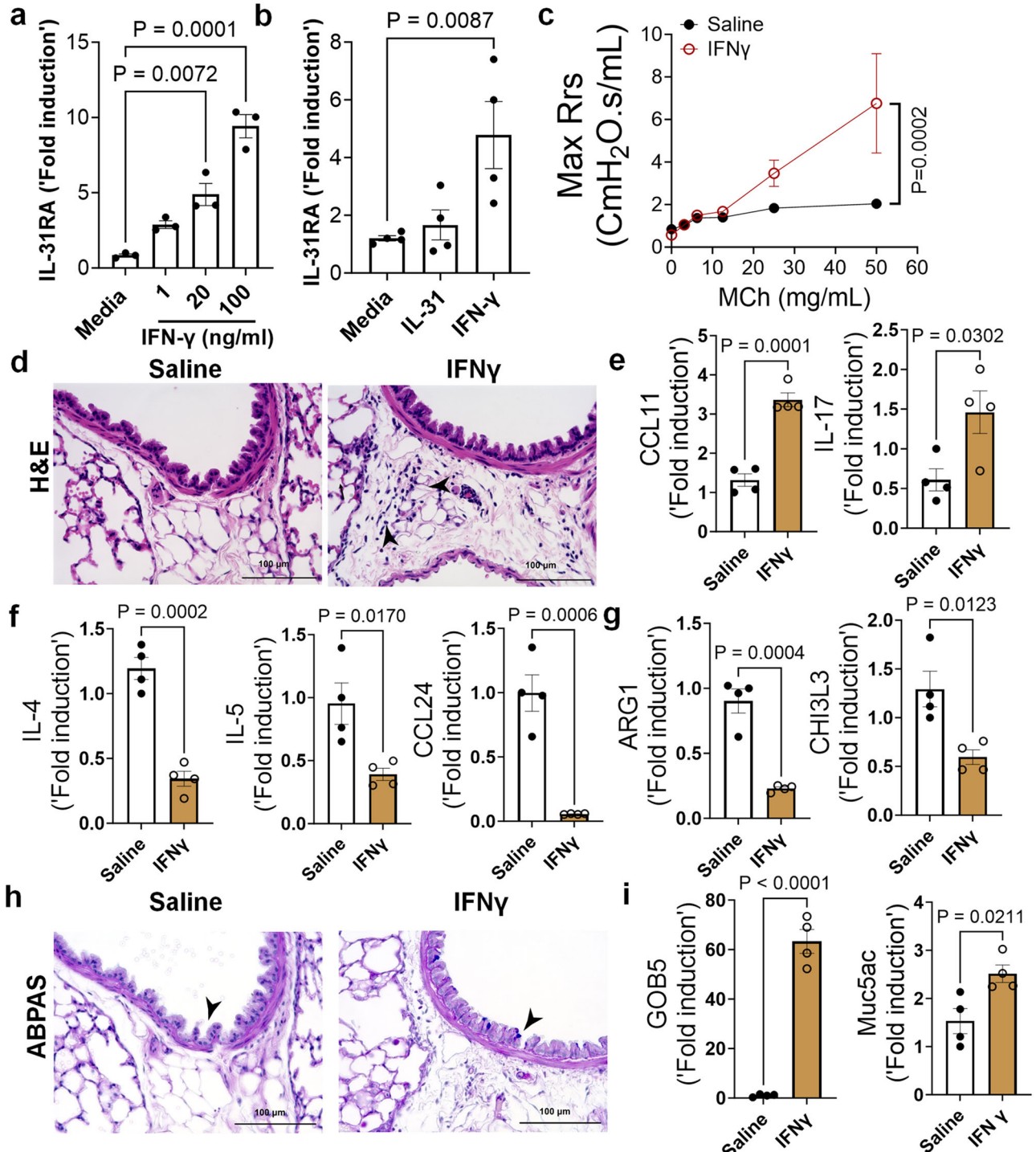

**Fig. 7 | IFNγ is a positive regulator of IL31RA expression, along with AHR, inflammation, and goblet cell hyperplasia. a** Quantification of IL-31RA transcript levels in mouse airway smooth muscle cells (ASMC) treated with increasing doses of IFNγ (*n* = 3) or media (*n* = 3) for 16 h. Data are shown as the mean ± SEM. Two-way ANOVA was used. **b** Quantification of IL-31RA transcript levels in human ASMCs treated with media, IL-31 (500 ng/mL) or IFNγ (50 ng/mL) for 16 h (*n* = 4). Data are shown as the mean ± SEM. Two-way ANOVA was used. **c** Wildtype mice were intratracheally treated with saline (*n* = 4) or IFNγ (5 μg) (*n* = 4) on days 0 and 6, and airway resistance with increasing doses of methacholine was measured on day 7. Data are shown as mean ± SEM. Two-way ANOVA was used. **d** Representative images of hematoxylin and eosin -stained lung sections from wildtype mice treated with saline (*n* = 4) or IFNγ (*n* = 4). Images were captured at ×20 magnification. Scale

bar, 100 μm. **e**, **f**, **g** Quantification of inflammatory chemokines (*CCL11* and *IL-17*), Th2 cytokines (*IL-4, IL-5, and CCL24*), and Th2 response-associated genes, including *ARG1* and *CHI3L3* in the lungs of wildtype mice treated with saline (*n* = 4) or IFNγ (*n* = 4). Data are shown as the mean ± SEM. Unpaired *t* test was used.

**h** Representative images of Alcian blue periodic acid-Schiff-stained lung sections from wild-type mice treated with saline (*n* = 4) or IFNγ (*n* = 4). Images were captured at ×20 magnification. Scale bar, 100 μm. **i** Quantification of mucus-associated gene transcripts, including *GOB5* and *MUC5AC*, in the lungs of wildtype mice treated with saline (*n* = 4) or IFNγ (*n* = 4). Data are shown as the mean ± SEM. Unpaired *t* test was used. At least two independent experiments produced similar results. Source data are provided as a Source Data file.

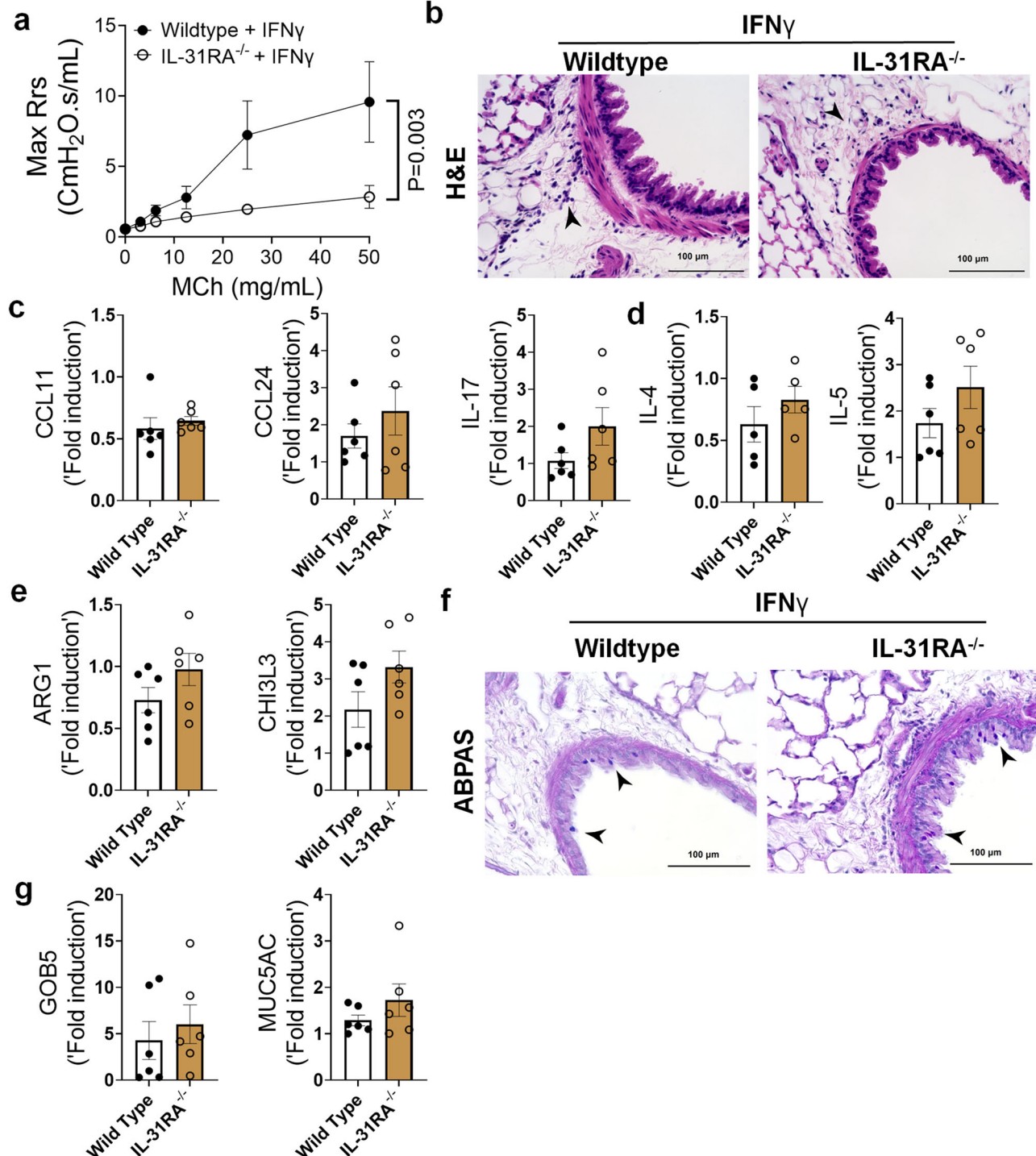

**Fig. 8 | Loss of IL-31RA is sufficient to attenuate IFNγ-induced AHR with no effect on inflammation and goblet cell hyperplasia. a** Wildtype (*n* = 9) and IL-31RA⁻/⁻ (*n* = 9) mice were treated intratracheally with IFNγ (5 μg) on days 0 and 6, and resistance was measured with increasing doses of MCh using Fexivent. Data are shown as mean ± SEM. Two-way ANOVA was used. **b** Representative images of hematoxylin and eosin-stained lung sections from wildtype (*n* = 6) and IL-31RA⁻/⁻ (*n* = 6) mice treated with IFNγ. Images were captured at ×20 magnification. Scale bar, 100 μm. **c, d, e** Quantification of inflammatory cytokines (*CCL11, CCL24* and *IL-17*), Th2 cytokines (*IL-4* and *IL-5*), and Th2 response-associated genes including *ARG1, and CHI3L3* in the lungs of wildtype (*n* = 6) and IL-31RA⁻/⁻ (*n* = 6) mice treated

with IFNγ. Data are shown as mean ± SEM. Unpaired *t* test was used, and no statistical signification observed between groups. **f** Representative images of Alcian blue periodic acid-Schiff-stained lung sections from wildtype (*n* = 6) and IL-31RA⁻/⁻ (*n* = 6) mice treated with IFNγ. Images were captured at ×20 magnification. Scale bar, 100 μm. **g** Quantification of mucus-associated genes including *GOB5* and *MUC5AC* transcript levels in the lungs of wildtype (*n* = 6) and IL-31RA⁻/⁻ (*n* = 6) mice treated with IFNγ. Data are shown as means ± SEM. Unpaired *t* test was used, and no statistical signification observed between groups. At least two independent experiments produced similar results. Source data are provided as a Source Data file.

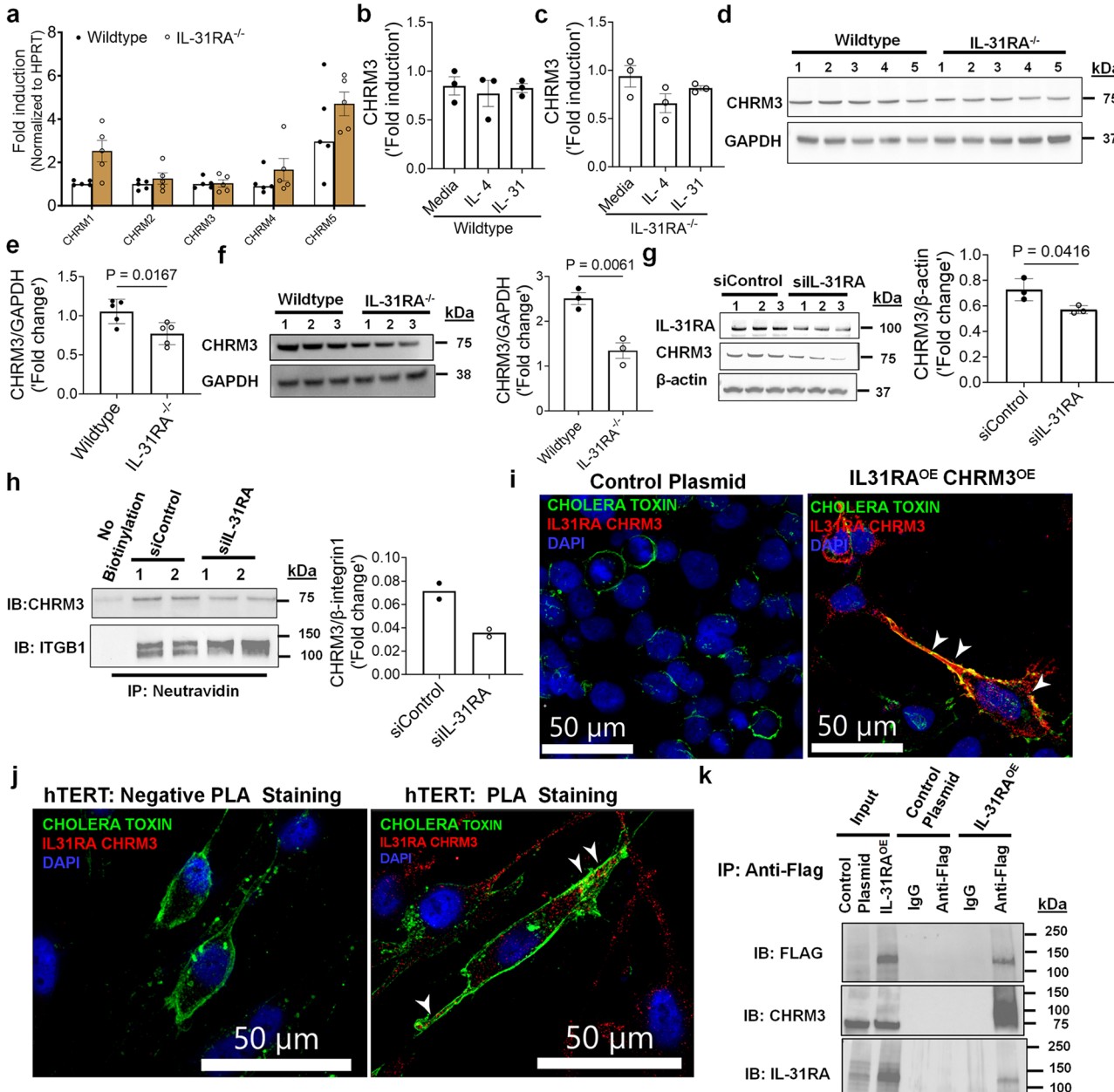

**Fig. 9 | The post-transcriptional regulation of CHRM3 expression by IL-31RA.**
**a** Quantification of CHRMs transcripts in the lungs of wildtype ($n = 5$) and IL-31RA$^{-/-}$ ($n = 5$) mice using RT-PCR. Data shown as mean ± SEM. Two-way ANOVA was used. **b, c** Quantification of CHRM3 transcripts in ASMC isolated from wildtype ($n = 3$) and IL-31RA$^{-/-}$ ($n = 3$) mice and treated with media, IL-4 (10 ng/mL) or IL-31 (500 ng/mL) for 16 h. Data shown as mean ± SEM. Two-way ANOVA was used. **d, e** The total lung lysates of HDM-treated wildtype ($n = 5$) and IL-31RA$^{-/-}$ ($n = 5$) mice were immunoblotted with antibodies against CHRM3 and GAPDH. Data are shown as mean ± SEM. Unpaired $t$ test was used. **f** ASMC isolated from wildtype ($n = 3$) and IL-31RA$^{-/-}$ ($n = 3$) mice were lysed and immunoblotted with antibodies against CHRM3 and GAPDH. Data are shown as mean ± SEM. Unpaired $t$ test was used.
**g** hTERT cells were transiently transfected with IL-31RA-specific ($n = 3$) or control siRNA ($n = 3$) for 72 h. Cell lysates were immunoblotted with antibodies against IL-31RA, CHRM3 and β-actin. Bar graph shows CHRM3 protein levels normalized to β-actin. Unpaired $t$ test was used. Data are shown as means ± SEM. **h** Cell surface proteins were biotinylated, and affinity purified using neutravidin to measure cell surface levels of IL-31RA and CHRM3 in hTERT cells transfected with IL31RA-specific

($n = 2$) or control siRNA ($n = 2$) for 72 h. hTERT cells without biotinylation were used as a negative control. CHRM3 protein levels were normalized to ITGB1. **i** HEK293 cell were transiently transfected with overexpressing plasmids for CHRM3 and IL-31RA or empty control plasmids for 48 h. Cells were treated with antibodies against IL-31RA and CHRM3 and the IL-31RA-CHRM3 complex formation was visualized using hybridization probes at an excitation λex 594 nm (Red). The plasma membrane was stained with cholera toxin subunit b conjugated with Alexa Fluor 488 (Green) and the nuclei with DAPI (Blue). The white arrowheads highlight colocalization between the cholera toxin and puncta of the IL-31RA-CHRM3 complex. Images were captured at ×40 magnification. Scale bar, 50 μm. **j** hTERT cells were treated with antibodies against IL-31RA and CHRM3 or isotype IgG (negative PLA staining). The IL-31RA-CHRM3 complex formation was visualized as described in Fig. 9i. **k** HEK293T cells transiently transfected with a control plasmid or FLAG-tagged IL-31RA$^{OE}$ plasmid for 72 h. Total cell lysates and eluted fractions were immunoblotted with anti-Flag, anti-IL31RA and anti-CHRM3 antibodies. At least two independent experiments produced similar results. Source data are provided as a Source Data file.

for surface biotinylation (Fig. 9h). The decrease in CHRM3 protein with no changes in its transcript levels in the absence of IL-31RA suggests a post-transcriptional regulation that may involve a physical interaction between IL-31RA and CHRM3 in ASMC.

To identify the complex formation between IL-31RA and CHRM3, we used in situ proximity ligation assay (PLA) in HEK293T cells overexpressing both IL-31RA and CHRM3. Importantly, we observed bright fluorescent signals corresponding to the IL-31RA-CHRM3 complex formation in HEK293T cells overexpressing IL-31RA and CHRM3 compared to control cells (Fig. 9i). By fluorescently labeling plasma membrane with cholera toxin, we demonstrated the colocalization of PLA puncta with the plasma membrane in HEK293T cells (Fig. 9i). Similarly, we observed the IL-31RA-CHRM3 complex formation in hTERT cells (Fig. 9j). The lack of PLA staining with isotype antibodies supports the specificity of the binding interactions between IL-31RA and CHRM3 (Fig. 9j). Next, we assessed the physical interaction between IL-31RA and CHRM3 using co-immunoprecipitation studies with HEK293T cells transfected with plasmid overexpressing C-terminal FLAG-tagged IL-31RA. When we immunoprecipitated FLAG-tagged IL-31RA from cell extracts with antibody against FLAG, endogenous CHRM3 protein was co-precipitated, as assessed by western blotting (Fig. 9k). Immunoprecipitation with control IgG or cell lysates from HEK293T cells transfected with control plasmid were used as controls. We could not coimmunoprecipitate endogenous CHRM3.

Based on the positive regulation of CHRM3 protein levels by IL-31RA, we hypothesized that IL-31RA augments CHRM3-driven calcium signaling. Therefore, we assessed the gain-of-function effects of IL-31RA on CHRM3-driven calcium signaling by carbachol in HEK293T cells. As expected, carbachol treatment of HEK293T cells that were transfected with control plasmids resulted in elevated intracellular calcium flux (Fig. 10a). Further, overexpression of IL-31RA augmented calcium release similar to the levels observed with CHRM3 overexpression. This increase in intracellular calcium release was further elevated in HEK293T cells that co-expressed both IL-31RA and CHRM3 which may suggest a cooperation between IL-31RA and CHRM3 to augment carbachol-induced calcium flux. Next, we assessed the effect of IL-31RA overexpression on another Gq-coupled GPCR agonist, bradykinin-induced calcium elevation. We observed limited or no effect of IL-31RA overexpression on calcium levels in HEK293 cells (Fig. 10b). Next, we assessed agonist-induced calcium responses in hTERT cells that retain CHRM3 expression for an extended number of passages. Overexpression of IL-31RA augmented carbachol-induced calcium release but not bradykinin- or serotonin-induced calcium elevation in hTERT cells (Fig. 10c, d, e). The change in the fluorescence values upon ionomycin stimulation was similar in cells transfected with control and IL-31RA overexpression plasmids (Fig. 10f). To determine the dependency of CHRM3 in IL-31RA-driven calcium elevation, we assessed carbachol-driven calcium elevation in hTERT cell line 72, which expresses very low levels of CHRM3[52,53]. In support of our hypothesis, we observed a significant attenuation of carbachol-driven calcium elevation in hTERT line 72 cells transfected with either the control plasmid or the IL-31RA overexpression plasmid (Fig. 10g). As expected, ionomycin-driven calcium levels remained similar and elevated in hTERT line 72 cells transfected with either the control plasmid or the IL-31RA overexpression plasmid (Fig. 10g). Similarly, siRNA-mediated knockdown of IL-31RA attenuated carbachol-induced calcium levels in hTERT cells that express high levels of CHRM3 (Fig. 10h). Also, we assessed IL-31 dependency in IL-31RA-driven calcium elevation in hTERT cells. Notably, treatment with IL-31 had no effect on carbachol-induced calcium levels in IL-31RA overexpressing hTERT cells (Fig. 10i). These data collectively support our hypothesis that IL-31RA-mediated effects on ASMC are specific to CHRM3.

Calcium-dependent phosphorylation of myosin light chain (MLC) is a terminal event in the contraction of ASMC [54]. Therefore, we measured carbachol-induced phosphorylation of MLC in hTERT cells and observed a significant decrease in carbachol-induced MLC phosphorylation with the knockdown of IL-31RA (Fig. 10j). To further demonstrate the positive regulation of MLC phosphorylation by the IL-31RA-CHRM3 axis, we measured carbachol-induced phosphorylation of MLC in ASMC isolated from wild-type and $IL\text{-}31RA^{-/-}$ mice. Consistent with reduced CHRM3 expression in IL-31RA deficient ASMC, we observed a significant decrease in carbachol-induced MLC phosphorylation in ASMC isolated from IL-31RA$^{-/-}$ mice compared to wild-type mice (Fig. 10k and Supplementary Fig. 11). Also, we measured MLC phosphorylation by serotonin and observed no change in serotonin-induced phosphorylation of MLC with the loss of IL-31RA in ASMC (Supplementary Fig. 12). To assess the positive regulation MLC phosphorylation by IL-31RA in SMC of other organs, we isolated intestinal SMC from wild-type and $IL\text{-}31RA^{-/-}$ mice and treated with carbachol to measure phosphorylation of MLC. Similar to ASMC, we observed a significant decrease in carbachol-induced phosphorylation of MLC with the loss of IL-31RA in intestinal SMC (Supplementary Fig. 13). Together, the above findings support the concept that the IL-31RA functions as a positive regulator of CHRM3 and associated canonical Gq signaling involving elevation of intracellular calcium to augment the contractility of ASMC (Supplementary Fig. 14).

## Discussion

In this study, we investigated the pathophysiological role of the IL-31/IL-31RA axis using two complementary mouse models of allergic asthma. Our findings suggest that IL-31RA is pivotal in the development of AHR. Further, our Th1/Th2 cytokine intratracheal instillation studies using wild-type and IL-31RA knockout mice revealed an uncoupling of AHR from other airway pathologies typically underlying asthma, including inflammation and goblet cell hyperplasia. Consistent with dominant role for IL-31RA but not IL-31 in asthma pathogenesis in these models, IL-31RA was elevated in wildtype mice exposed to allergens and in human lung tissues obtained from asthmatics. Administration of IL-31 had (1) limited or no effect on MCh-induced AHR, inflammation, and goblet cell hyperplasia and (2) no significant changes in the contraction of airways in the PCLS or collagen gel-embedded ASMCs.

Our findings suggest that the expression of IL-31RA is not essential for Th1 and Th2 cytokine-induced inflammation and goblet cell hyperplasia. However, this apparently limited role of IL-31RA is consistent with a recently published study in which neutralization of IL-31 had no effect on airway inflammation in wild-type mice that were sensitized and challenged with ovalbumin[55]. Moreover, a study by Neuper et al. demonstrated an elevated expression of IL-31RA but not IL-31 in the lungs of mice challenged with the timothy grass pollen allergen[56]. The role of IL-31RA has been investigated in other disease models, including pulmonary fibrosis and systemic sclerosis. In support of our findings, the loss of IL-31RA had a limited effect on the expression of Th2 cytokines and inflammation but resulted in significant improvement in lung function and pulmonary fibrosis[18,30,57]. Upon repetitive injury with bleomycin, mice deficient in IL-31RA were protected from worsening lung function compared with wild-type mice[30]. However, studies using SEA-induced pulmonary injury model demonstrated elevated granulomatous inflammation and Th2 responses in the absence of IL-31RA[40]. Similarly, studies using the gastrointestinal helminth *Trichuris muris* infection model support elevated Th2 responses and accelerated expulsion of *Trichuris* with significantly decreased worm burdens in the absence of IL-31RA compared with their wild-type counterparts[40,55,56,58]. Therefore, we used two different models of allergic asthma, i.e., HDM and SEA. In both HDM and SEA models, we observed no significant differences in allergen-driven Th2 responses between wild-type and IL-31RA deficient mice but observed a significant improvement in AHR. Also, our findings demonstrated that the loss of IL-31RA is sufficient to attenuate both IFNγ- and IL-4/IL-13-induced AHR without inhibiting the infiltration of inflammatory cells in the lung. These findings suggest that

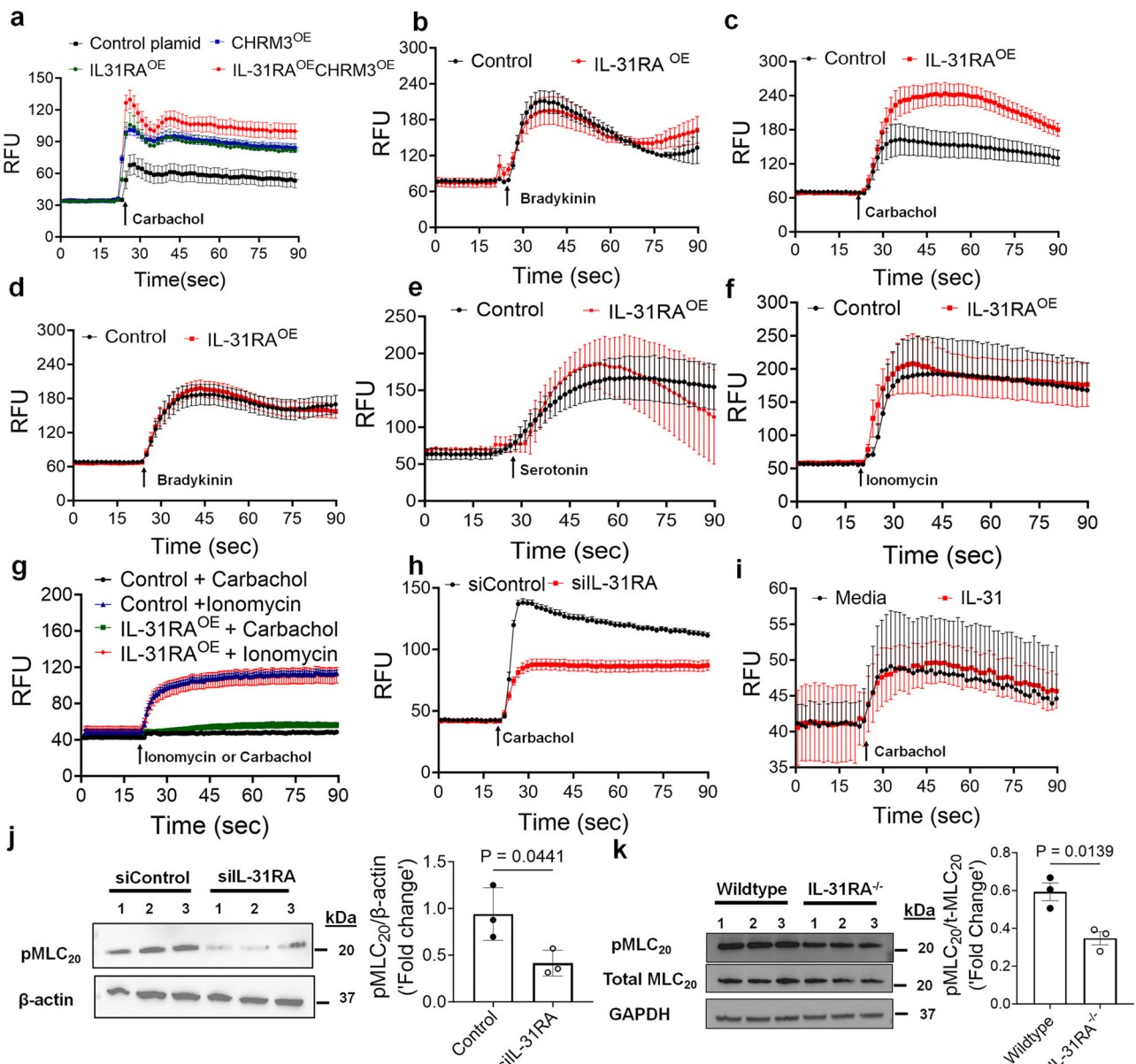

**Fig. 10 | The positive regulation of agonist-induced calcium elevation and MLC phosphorylation by the IL-31RA-CHRM3 axis. a** Intracellular calcium levels were measured using Fluo-4 AM dye in HEK293T cells transiently transfected with control or overexpressing plasmids for CHRM3 and/or IL-31RA plasmids for 72 h and treated with 10 μM carbachol ($n = 4$). **b** Intracellular calcium levels were measured using Fluo-4 AM dye in HEK293T cells transiently transfected with control or overexpressing plasmid for IL-31RA for 72 h and treated with 50 μM bradykinin ($n = 3$). **c, d, e, f** Intracellular calcium levels were measured using Fluo-4 AM dye in hTERT cells transiently transfected with control or overexpressing plasmid for IL-31RA for 72 h and cells treated with 10 μM carbachol ($n = 8$), 10 μM bradykinin ($n = 8$), 100 μM serotonin ($n = 4$) or 10 μM ionomycin ($n = 4$). **g** hTERT cell line 72 (CHRM3$^{LOW}$) was transiently transfected with control or IL-31RA overexpression plasmids for 72 h and intracellular calcium levels were measured in cells treated with 10 μM carbachol ($n = 5$) or 10 μM ionomycin ($n = 3$) using Fluo 4 AM dye. Data are shown as means ± SEM. **h** Intracellular calcium levels were measured using Fluo-

4 AM dye in hTERT cells transiently transfected with IL-31RA-specific or control siRNA for 72 h and treated with 10 μM carbachol ($n = 8$). **i** hTERT cells were transiently transfected with IL-31RA overexpressing plasmid for 48 h and treated with IL-31 (500 ng/mL) or media for another 24 h. Media or IL-31 treated cells were stimulated with 10 μM carbachol ($n = 4$) and intracellular calcium levels were measured using Fluo-4 AM dye. Data are shown as means ± SEM. **j** hTERT cells were transiently transfected with IL-31RA-specific or control siRNA for 72 h and treated with 10 μM carbachol ($n = 3$) for 10 min. Cell lysates were immunoblotted with antibodies against phospho-MLC, and β-actin. Data are shown as means ± SEM. Unpaired *t* test was used. **k** ASMC isolated from wildtype and IL-31RA$^{-/-}$ mice were treated with 10 μM carbachol ($n = 3$) for 10 min and cell lysates were immunoblotted with antibodies against phospho-MLC, total-MLC and GAPDH. Data are shown as means ± SEM. Unpaired *t* test was used. At least two independent experiments produced similar results. Source data are provided as a Source Data file.

IL-31RA-driven AHR could be downstream of both Th1 and Th2 cytokines. Using four different animal models and IL-31RA knockout mice, our findings robustly demonstrate the role of IL-31RA in the development of allergen-induced AHR despite lack an effect on airway inflammation. Future studies are needed to assess the effect of smooth muscle cell-specific IL-31RA deficiency on AHR and inflammation using

both helminth infection and allergic asthma models. These findings are consistent with increases in IL-31RA levels in human lung tissues obtained from patients with asthma. Using IL-31RA as a biomarker or its correlation to asthma severity need to be further investigated. Very interestingly, we did not find a significant role for IL-31, a Th2 cytokine in the development of AHR. Considering IL-31 is the only known

activator of IL-31RA, future studies are warranted to determine the molecular identity of mediator(s) that activate IL-31RA to alter the contractile functions of ASMC and AHR in asthma. These studies should be complemented by the assessment of the potential role of both soluble and membrane associated isoforms of IL-31RA in augmenting the contractile function of ASMC and AHR in asthma[46].

IFNγ is a potent Th1 cytokine primarily produced by Th1 T cells that is known to induce AHR in asthma[48,59,60]. Patients with severe asthma who respond poorly to corticosteroid therapy have been shown to have dominant Th1 responses, including elevated IFNγ in their lungs[48,61]. In vivo studies using animal models of severe asthma have suggested that IFNγ, but not IL-17, is responsible for heightened AHR[48]. In another study, the blockade of IFNγ attenuated Th1-cell-induced AHR, and this improvement in AHR appeared to be independent of neutrophils[59]. Similarly, multiple preclinical and clinical studies using antibody-mediated blockade of IL-4/IL-13 signaling have demonstrated the pathogenic role of Th2 cytokines in the development of AHR in allergic asthma[11,44,62]. However, the mechanisms underlying Th1/Th2 cytokine-driven AHR remain unclear. Although the hypothesis tested in this study was straightforward, it was surprising that IL-31RA, but not the Th2 T cell-derived cytokine, IL-31 was responsible for inducing AHR in allergic asthma. Nonetheless, we demonstrated that both Th1 and Th2 cytokines upregulated IL-31RA in ASMC and in the two complementary mouse models of allergic asthma. Our findings on the effect of IFNγ-induced IL-31RA expression are supported by previous studies in which IFNγ was shown to induce the expression of IL-31RA in dendritic cells, lung epithelial cells, and fibroblasts[19,24,63,64]. A recent study by Kobayashi et al. also demonstrated the role of IFNγ in the induction of AHR without significantly affecting airway inflammation using intranasal administration of IFNγ in mice[47]. However, future studies are required to determine mechanisms that uncouple AHR from inflammation and goblet cell hyperplasia in asthma.

We also show that IL-31RA functions as a positive regulator of CHRM3-driven calcium signaling and contractility of ASMC. This is the study, to our knowledge to show that the IL-31RA-CHRM3 axis enhances agonist-induced canonical Gq-calcium signaling to promote the phosphorylation of MLC in ASMC. CHRM3 is a key muscarinic receptor expressed by smooth muscle cells in the airways and plays a major role in the contractility of smooth muscle cells in response to muscarinic ligands such as acetylcholine (endogenous ligand released by parasympathetic nerve terminals) and methacholine (clinically used muscarinic ligand to assess airway responsiveness)[50,54]. CHRM3 is a member of class A GPCR that can mediate ASMC contraction through both calcium-dependent and calcium-independent mechanisms[49,54]. The calcium-dependent smooth muscle cell contractility is dependent on the activation of a subunit of Gq and production of inositol 1,4,5-triphosphate by phospholipase C[65]. Our findings using both the loss-of-function and gain-of-function approaches suggest that IL-31RA is a positive regulator of CHRM3 protein levels to augment carbachol driven calcium elevation and phosphorylation of MLC in ASMC. Using other known Gq-coupled GPCR agonists such as bradykinin and serotonin, and ionomycin, we demonstrate that IL-31RA-mediated effects in ASMC are specific to CHRM3. Further, findings of the current study suggest a significant increase in cell surface levels of CHRM3 by IL-31RA with limited transcriptional changes in ASMC. We observed a possible physical interaction between IL-31RA and CHRM3 that may contribute to enhanced calcium signaling and contractility of ASMC through stabilizing CHRM3 protein. It is possible that other mechanisms that involve stabilizing CHRM3 or other proteins in macromolecular complexes involved in calcium-dependent and calcium-independent ASMC contraction may be involved[49,54]. How these observations, including the physical interaction between IL-31RA and CHRM3 are linked to elevated CHRM3-driven signaling remains unknown. Further studies are required to determine the validity of the hypothesis that IL-31RA stabilizes CHRM3 to augment intracellular calcium signaling in the contraction of ASMC. Previous studies have demonstrated the role of chaperone proteins in cell surface translocation and stability of GPCRs including muscarinic receptors[66,67]. It is possible that IL-31RA acts as a chaperone to regulate CHRM3 trafficking in ASM cells. Also, it is not known whether the effects of IL-31RA are limited to CHRM3, or if it can alter CHRM2 expression and signaling to modulate the contractility of ASMC and AHR. Therefore, future studies are needed to explicitly explore these mechanisms. Nevertheless, given the robust increases in CHRM3-driven calcium elevation, phosphorylation of MLC, and ASMC contraction, it is easy to envision the scope of the IL-31RA-CHRM3 axis in inducing AHR in asthma. Our findings provide evidence of the potential therapeutic benefits of inhibiting the IL-31RA-CHRM3 axis downstream of both Th1 and Th2 cytokines, to attenuate AHR. Disrupting regulatory mechanisms by which IL-31RA alters CHRM3 expression and Gq-PLC activation that impact contractility function of SMC is an attractive avenue for antiasthma drug development. Considering the contractile effects of IL-31RA in SMC isolated from multiple organs, identifying inhibitors that can mitigate IL-31RA-driven effects in SMC will have a therapeutic value beyond asthma. Dupilumab therapy that inhibits both IL-4 and IL-13-driven signaling significantly improved lung function and reduced severe exacerbations in patients with uncontrolled asthma[68]. However, there is still a gap in our understanding of mechanisms underlying AHR and a potential to combine these established therapies with inhibitors of the IL-31RA-CHRM3 axis to achieve effective inhibition of AHR, inflammation and goblet cell hyperplasia in asthma.

In summary, we have described the role of IL-31RA in the induction of AHR, with limited effects on airway inflammation and goblet cell hyperplasia, and this role is independent of its ligand IL-31. Importantly, we identified a key mechanism underlying ASMC contractility that involves IL-31RA-dependent increases in CHRM3 expression, calcium elevation and phosphorylation of MLC. Together, these results suggest an important role for IL-31RA in the regulation of AHR, and identifying the molecular events underlying IL-31RA-driven AHR may lead to the development of therapeutic approaches against AHR in allergic asthma.

## Methods
### Human samples
Asthma ($n = 8$; 6 M and 2 F) and normal lung samples ($n = 8$; 5 M and 3 F) were obtained with the assistance of the Translational Pulmonary Science Center (TPSC), University of Cincinnati Medical Center. The local University of Cincinnati institutional review board (IRB # 2013-8157) approved the protocol for human sample collection and laboratory analysis. Human samples are collected with informed consent and de-identified as per protocol.

### Animals
Wildtype (JAX Strain # 000664) and IL-31RA knockout (*IL-31RA*$^{-/-}$) mice of C57BL/6 background were used for all the experiments[19,30]. Age-matched male and female mice of 12–16 weeks of age were used for experiments with at least two repeats. A functional loss of IL-31RA signaling has been induced in these mice through targeted deletion of 4–6 exons in the mouse genome, which encodes the cytokine-binding domain 2 of IL-31RA. All mice were housed under specific pathogen-free conditions at the Cincinnati Children's Hospital Medical Center, a medical facility approved by American Association for the Accreditation of Laboratory Animal Care. All experimental procedures were approved by the Animal Care and Use Committee of Cincinnati Children's Hospital Medical Center.

### Antibodies, primers, and siRNA
The details of antibodies used are provided in Supplementary Table 1. The sequence details of RT-PCR primers and siRNA used are provided in Supplementary Table 2.

## HDM- and SEA-induced allergic asthma mouse models

Mice were challenged with HDM extracts (*Dermatophagoides pteronyssinus*, Greer laboratories Inc), a clinically important and commonly used allergen to induce asthma without the need for an adjuvant. Mice were sensitized twice with 200 μg of HDM in 200 μL of phosphate-buffered saline (PBS) through intraperitoneal injections on days 0 and 7. Mice were further challenged on days 14 and 16 through intratracheal instillation of 50 μg of HDM in 50 μL PBS. Mice were anesthetized with ketamine and xylazine to administer the allergen to the airways. Twenty-four hours following the final challenge with HDM, mice were anesthetized to assess the AHR using flexiVent (Scireq), and tissues were collected for biochemical and histological analyses. Control mice were sensitized with HDM and challenged with PBS.

The SEA prepared from *Schistosoma mansoni* helminth was used to induce AHR and inflammation in mice. SEA has been used for decades as an antigen or allergen owing to its robust induction of type 2 immune responses in animal models. Previous studies have described SEA-induced asthma model[21,31]. Briefly, mice were sensitized with 10 μg of SEA in 200 μL by intraperitoneal injection on days 0 and 14. Two weeks later, mice were anesthetized with ketamine and xylazine, and then challenged by intratracheal instillation with 10 μg of SEA in 50 μL of PBS, twice on days 28 and 31. The day following the last challenge, the mice were euthanized, and samples were collected for further analysis.

## Cytokine treatments in vivo

To evaluate the specific role of cytokines in the induction of AHR and inflammation, wild-type and *IL-31RA*[-/-] mice were intratracheally administered with recombinant mouse cytokines[11,13]. In brief, mice were anesthetized by an intraperitoneal injection of ketamine/xylazine, and treated with rIL-31 (10 μg R&D Systems), rIFNγ (5 μg, R&D Systems), or rIL-13 (5 μg, R&D Systems) in 50 μL of saline intratracheally instilled on days 0 and 6. Twenty-four hours after the last challenge, mice were euthanized, airway resistance was measured using FlexiVent, and the lungs were collected to assess histological changes and gene expression analysis using reverse transcriptase PCR (RT-PCR). Control mice were treated with a saline solution.

## Preparation of BAL cells

Twenty-four hours after the last challenge with the allergen, the mice were anesthetized, and the lungs were lavaged twice with 1 mL of ice-cold and sterile PBS using a 1 mL insulin syringe through a tracheal catheter. BAL fluid (BALF) was centrifuged for 5 min at $250 \times g$ at 4 °C. Cell pellets containing BAL cells were pooled and resuspended in 500 μL of PBS. The supernatant from the first BALF aspiration was stored at −80 °C for cytokines measurement using LEGENDplex™. Total cell counts were determined using an automated cell counter (Thermo Fisher Scientific, Waltham, MA, USA). BAL cells were used for cytospin preparation and stained with the Diff-Quick staining kit (Thermo Fisher). Differential cell counts were determined using morphological criteria under a microscope.

## BALF cytokine measurements

The concentration of cytokines in BALF from wild-type and IL31RA[-/-] mice challenged with HDM or saline was determined using a custom LEGENDplex™ mouse Panel 761 kit, designed to detect IFNγ, IL-4, IL-5, IL-6, IL-10, IL-13, and TNF-α cytokines, following the manufacturer's instructions. Signal intensities from standards and samples were acquired on a BD FACS LSRFortessa flow cytometer, and data were analyzed using LEGENDplex V8.0 Data analysis Software (BioLegend) (Supplementary Fig. 15).

## Histology

Mice were euthanized 24 h after the last allergen (HDM or SEA) challenge or after the last cytokine instillation. The lungs were inflated, fixed in 10% neutral formalin, and collected for histological analysis. Paraffin-embedded 5 μm sections of the left lobe were stained with H&E to examine associated airway inflammation and peribronchial cell infiltration. ABPAS (Alcian blue periodic acid Schiff) staining was used to assess goblet cell hyperplasia and mucus hypersecretion. Goblet cells were further counted on lung sections and expressed as the percentage of ABPAS-positive cells of the total nucleated cells of the airway epithelium, using Metamorph imaging software (Molecular Devices). At least 10 images per mouse were used to count goblet and total epithelial cells[64]. Images were captured at 40x magnification using a Keyence BZ-X microscope with a focus on the airway areas to quantify the ABPAS-positive cells.

## Immunohistochemistry

The lung sections were incubated with primary and HRP-conjugated secondary antibodies (anti-human IL31RA, 15 μg/mL, R&D Systems; anti-mouse IL-31RA, 2 μg/ml, Bristol Myers Squibb Inc; IgG isotype control, 15 μg/mL, R&D Systems) and developed using protocols provided by Vector Laboratories. Images were collected and analyzed using a Keyence BZ-X microscope.

## Lung function measurements

Methacholine (MCh)-induced AHR was measured using FlexiVent system (SCIREQ, Montreal, QC, Canada). Twenty-four hours after the last challenge, the mice were anesthetized using a ketamine/xylazine mix. Tracheotomy was performed, and the trachea was cannulated using a stainless catheter and connected to the FlexiVent machine for airway resistance measurements. Increasing doses of MCh (3–50 mg/mL; Sigma Aldrich) were aerosolized using an ultrasonic nebulizer connected to the FlexiVent system. and administrated following the manufacturer instructions after the baseline readings are stabilized. Following deep inflation, lung function measurements were recorded, and presented as the maximum resistance recorded for increasing MCh doses. We used whole body plethysmography (Buxco Electronics) to measure AHR in wild-type and IL-31RA[-/-] mice in response to increasing doses of inhaled MCh (3–50 mg/mL) as per manufacturer instructions.

## PCLS model

Eight mice of 10 weeks of age were euthanized with a lethal dose of sodium pentobarbital. The lungs were inflated with warm low melting agarose (Sigma) at 37 °C through a tracheal cannula, followed by 0.2 mL of air to flush the agarose. The left lung and apical and basal right lobes were used to generate PCLS. Lungs were sliced into 300 μm section in 4 °C cold HBSS without calcium using a VF-310Z vibratome (Pecisionary Instruments, Winchester, MA, USA). PCLS were collected in DMEM media without serum, and the media were changed at least four times with intermittent shaking to remove residual agarose and incubated in a humidified incubator with 5% $CO_2$ at 37 °C. On the following day, the media was replaced with fresh media without serum for the same-day experiment. For cytokine treatments, PCLS were treated with IL-13 (50 ng/mL, R&D Systems) or IL-31 (500 ng/mL, R&D Systems) for 24 h in a low-serum media (1% FBS) prior to the contraction assay. PCLS were mounted and images were obtained at baseline. After obtaining images, the slices were stimulated with increasing doses of MCh. Changes in the airway lumen area were recorded in a time-lapse setting with images captured every 5 s for 5 min following each dose of MCh using a 10x objective on a temperature-controlled Nikon inverted microscope at 37 °C with 5% $CO_2$ (NIKON Ti2 widefield, Japan). Airway areas were analyzed using NIKON NIS-Elements analysis software, and changes were expressed as the percentage change compared to the initial baseline area.

## RNA isolation and real-time PCR

Lung tissues were dissected and homogenized using TRIzol solution (Life Technologies) using beads and a high-speed homogenizer

(Thermo Fisher). RNA was isolated from lung tissues, and ASMC using an RNAeasy mini kit (QIAGEN) following the manufacturer's instructions. cDNA was synthesized using Superscript III (ThermoFisher), and real-time PCR was performed using SYBR Select Master Mix (Bio-Rad) and a CFX384 Touch Real-Time PCR instrument (Bio-Rad). Data were analyzed using CFX Maestro software version 4.0. Target gene transcripts were normalized using hypoxanthine-guanine phosphoribosyl transferase (HPRT, for murine cells and tissues) or β-actin (for human cells) as housekeeping genes. The list of primers used to measure the transcript levels of the genes of interest can be found in the supplementary information.

### Preparation of SMC and collagen gel contraction assays
Airway and intestinal SMCs were prepared by enzymatic digestion and culturing of trachea and intestinal tissues, respectively. Briefly, murine trachea and intestines were collected, and epithelium was removed by scraping and washing in HBSS. Tissues were minced and digested enzymatically with collagenase type IV and dispase II in HAM's F12 media with no serum. A single cell suspension of digested tissues was seeded on a 100 mm petri dish ($n = 4$/dish) using Hams/F12 culture media supplemented with 10% FBS and antibiotics. Spindle-shaped cells were allowed to grow to 80% confluence, and cells from passages 1–2 were used for signaling studies and the collagen gel contraction assay. ASMCs were seeded into rat tail collagen gel (ThermoFisher Scientific) matrices at 200,000 cells/600 μL gel volume. To determine the effect of cytokines on ASMC contraction, cells were treated with cytokines (rIL-31 500 ng/mL or rIL-13, 50 ng/mL) or media for 48 h prior to embedding them in collagen gel. Collagen gels were prepared using cells treated with cytokines or media and grown under similar conditions. The collagen gels were detached from the walls, and images were captured (0–48 h) using a stereo microscope. The area of collagen gel containing ASMC was measured using ImageJ software and the percentage change from the baseline in response to agonist stimulation was calculated.

### Western blotting
Cell lysates were prepared using the RIPA lysis buffer containing protease and phosphatase inhibitors. After SDS−PAGE separation, proteins were transferred to nitrocellulose membrane and incubated overnight with primary antibodies against CHRM3 (1:1000, Abcam), phosphoMLC (1:1000, Cell Signaling Technology), totalMLC (1:1000, Cell Signaling Technology), hIL-31RA (1:100, R&D Systems), ITGB1 (1:1000, Cell Signaling Technology) and GAPDH (1:2000, Bethyl Laboratories) and followed by detection with HRP conjugated secondary antibodies. Proteins were quantified by normalizing to GAPDH levels using the volume integration function of the BIORAD imager software. The samples for western blots were derived from the same experiment and processed in parallel.

### Calcium flux assay
Human TERT cells, human TERT line 72 or HEK293T cells were grown in F12 media with 1% ITS, penicillin/streptomycin, NaOH, CaCl2, HEPES and L-Glutamine for 72 h. On day 3, cells were loaded with Fluo-4 AM (5 μM) (Invitrogen) for one hour in HBSS and stimulated with agonists to measure changes in the fluorescence in real time for 90 sec using a Flex Station III (Molecular Devices).

### In situ proximity ligation assay (PLA)
hTERT cells or HEK293T cells were transiently transfected with over-expressing plasmids for IL-31RA and CHRM3 or control empty plasmids (Origene) using Lipofectamine 3000 and cultured for 48 h. Then, cells were fixed for 5 min in ice cold methanol at 4 °C and incubated with Duolink blocking solution (Sigma) for 1 h at room temperature followed by overnight incubation with primary antibodies against IL-31RA (15 μg/mL, R&D Systems) and CHRM3 (1:500, Abcam). The

incubation was followed by three 5 min washes and incubation with PLA probes anti-goat PLUS (DUO92003) and anti-rabbit MINUS (DUO92005) for 1 hr at 37 °C as recommended in the Duolink PLA kit (Sigma). After three washes, the cells were incubated with the hybridization solution containing DNA ligase for 30 min at 37 °C. Cells were washed and incubated with the amplification-polymerase at 37 °C for 100 min, washed three times and mounted with DAPI mounting medium. To identify the membrane localization of in situ PLA signals, the cells were counterstained with cholera toxin conjugated with Alexa Fluor 488 (1:200, Thermo Fisher). Fluorescence images were acquired using a Nikon A1R confocal laser scanning microscope and analyzed using Imaris image analysis software.

### Immunoprecipitation studies
HEK293T cells were transiently transfected with overexpressing plasmids for IL-31RA (Origene) or empty control plasmid using Lipofectamine 3000 and cultured for 72 h. Cell lysates were prepared in RIPA buffer supplemented with protease inhibitor cocktail and incubated with agarose A/G beads conjugated with anti-FLAG or control IgG antibodies. Immunoprecipitants were collected by centrifugation, washed three times with PBS, eluted with low pH glycine buffer and subjected to SDS-PAGE with Laemmli sample buffer. Membranes were immunoblotted with anti-Flag, anti-IL31RA and anti-CHRM3 antibodies followed by detection with HRP conjugated secondary antibodies and visualized with ECL on a ChemiDOC Touch imaging system (Bio-Rad).

### Cell surface protein biotinylation assay
hTERT cells were transiently transfected with control or IL-31RA-specific siRNA or HEK293 cells with control or IL-31RA overexpression plasmid using lipofectamine 3000. After 72 h of transfection, cell surface proteins were biotinylated using Surface Biotinylation kit from Pierce™ according to the protocol and instructions from the manufacturer. Biotinylated membrane proteins were affinity purified from cell lysates using NeutrAvidin™ Agarose slurry. Cells without biotinylation used as a negative control. The biotinylated membrane proteins were eluted with elution buffer containing 30 mM DTT and analyzed using western blots.

### Statistical analysis
Data were analyzed using Prism (version 9; GraphPad, San Diego, CA, USA). Quantitative data (mean ± SEM) are presented as bar graphs with individual dots. Data were considered statistically significant when the $p$ values were less than 0.05. Unpaired $t$ test was used to compare the means between the two groups. For multiple comparisons, one-way ANOVA with post hoc Tukey's test or 2-way ANOVA with post hoc Sidak's test was used.

### Reporting summary
Further information on research design is available in the Nature Portfolio Reporting Summary linked to this article.

## Data availability
All other data are available in the article and its supplementary information files or from the corresponding author upon request. Source data are provided with this paper.

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

## Acknowledgements
We thank Dr. Stacey R. Dillon (ZymoGenetics, a Bristol-Myers Squibb Company, Seattle, WA) for providing antibody to mouse IL-31RA and IL-31RA[-/-] mice. This research was supported by NIH/NHLBI grants R01HL157176 and R01HL134801 (to SKM)., and Department of Biotechnology, Government of India (to SA).

## Author contributions
Conceptualization: S.K.M.. Methodology: S.K.M., D.J.K.Y., S.V.A., S.Y., D.A.D.. Investigation: S.K.M., D.J.K.Y., S.V.A., D.A.D.. Visualization: S.K.M., D.J.K.Y., S.V.A.. Supervision: S.K.M., G.B.R., F.X.M.. Writing—original draft: S.K.M., D.J.K.Y., S.V.A.. Writing—review & editing: S.K.M., D.J.K.Y., S.V.A., G.B.R., F.X.M., D.A.D..

## Competing interests
The authors declare no competing interests.
