## [Peer Review File · Nature Communications]

REVIEWER COMMENTS

Reviewer #1 (expert in airway hyperresponsiveness and airway smooth muscle cells):

Summary and Novelty: Using two murine models of allergen-induced AHR, the authors determined that IL-31RA modulates baseline methacholine (Mch)-induced airway smooth muscle (ASM) shortening and calcium mobilization and attenuates allergen-induced AHR by decreasing ASM responsiveness that is independent of allergen-induced airway inflammation and goblet cell hyperplasia. The effects of IL-31RA on Mch responsiveness on ASM appear independent of IL-31 activation of IL-31RA but can be mediated by IL-4/IL-13- or IFN- γ -induced expression of IL-31RA on ASM that in turn enhances expression of ASM M3R expression and Mch-mediated calcium responses. The observation that IL-31RA modulates Mch-induced shortening, contraction, and calcium responses in murine ASM is novel. However, the recognition that allergen-induced AHR can be uncoupled from airway inflammation and goblet cell hyperplasia has been well established. The MS is well-written and the results support some of the conclusions; significant concerns should be addressed that would enhance the quality of the MS.

Major Concerns:

1. The findings exclusively using murine cells and models limit the physiological and pharmacological relevance to humans or to asthma. Human ASM cells are profoundly different from murine ASM; validation of the IL-31RA findings in human-derived ASM is critically important to understand the significance of their results.
2. Since non-specific AHR is a hallmark of asthma and since human ASM responds to many contractile agonists, the investigators should address whether their murine ASM findings are exclusively due to IL-31-M3R modulation. In other words, does IL-31 also modulate other agonist responses in murine ASM. The authors could use serotonin as an agonist; this question is critically important to understand how IL-31 RA mediates its effects.
3. It is surprising that IL-31 alone had little effect on ASM excitation-contraction (EC) coupling. The global knockout of IL-31RA can have myriad effects that could contribute to the phenotype observed. This is a significant limitation of the studies. Does IL-31RA serve as a scaffold protein that enables G β q signaling? Such a possibility was not addressed.
4. In the discussion, the investigators posit that targeting IL-31RA could mitigate Mch-induced AHR. How would the investigators modulate IL-31RA expression/activation since IL-31 alone has no effect? Certainly, the elimination of IL-31 would not be a viable approach and accordingly blockade of the receptor would not be useful.
5. In Figure 1.D., the KD of IL-31RA at baseline attenuates Mch responses in the absence of HDM. Are these data significant compared to WT/saline? This is important and suggests that the global KO of IL-31 impairs ASM contractility. Indeed, does the KO of IL-31RA impair ALL smooth muscle responses to activation of the M3R for example in the gut or vasculature? This is relevant if as the investigators posit, IL-31RA can be a therapeutic target.

Minor Concerns:

1. The title is inaccurate since the authors do not study human cells or tissue associated with asthma. The authors study allergen-induced AHR and airway inflammation.
2. The introduction is too long and rambling; there should be more focus.
3. In the abstract, the authors state that both Th1 and Th2 cytokines induce asthma, which is incorrect. These cytokines are associated with asthma but there is no evidence of causation.

Reviewer #2 (expert in asthma and lung inflammation):

In the manuscript by Akkenepally et al., the authors show that loss of IL-31RA attenuated methacholine induced AHR in HDM and SEA models, without affecting Th2 cytokine expression, mucus production, or recruitment of inflammatory immune cells. Interestingly, IL-31 administration did not have a significant effect on airway contractility. However, loss of IL-31RA attenuated IL-13 induced ASM hyperreactivity without affecting IL-13's ability to induce goblet cell hyperplasia. Similarly, loss of IL-31RA attenuated IFN γ induced AHR without affecting inflammation or mucus production. IL-31RA deficiency led to a decrease in CHRM3 protein expression without changes in transcript level. Furthermore, IL-31RA expression in HEK293T cells was able to augment carbachol-induced myosin light chain phosphorylation, calcium influx, possibly through interaction with CHRM3. Overall, this work identifies a novel mechanism through which IL-31RA selectively mediates AHR uncoupled from airway inflammation. However, there are specific issues that if addressed would significantly support the authors conclusions.

Major comments:

1. Strain specificity. The authors use B6 mice strain and previous studies have shown that AHR differences can occur in strain-specific manners. To more broadly accept the conclusions regarding the important role of IL-31R selectively in AHR, IL-31/31R blocking studies in another strain of mice (e.g. Balb) in an HDM model would strengthen the overall work.
2. The current manuscript demonstrates that IL-31R is important for the development of AHR but a critical question therapeutically is whether blockade of IL-31/31R after lung inflammation is established will also show the same reduction in AHR. IL-31/R blocking studies could address this,.
3. The in vitro studies showing a potential interaction between CHRM3 and IL-31R to regulate ASM calcium influx and MLC phosphorylation are interesting. Gain of function studies in HEK293T cells should be complemented by loss of function through siRNA knockdown or genetic knockout of CHRM3 to see if this attenuates IL-31RA intracellular calcium flux. Further, less artificial studies in ASM after Th2 cytokine upregulation of IL-31R to assess CHRM3-IL-31R interactions would be more relevant to the in vivo data.

4. The manuscript shows mostly transcript levels of key endpoints (cytokines, IL-31R expression) without corroborative protein expression. Further, tissue protein staining of IL-31R at baseline and after T2 HDM model could support the more selective role of IL-31R in ASM function compared with function in other cell types. Overall, demonstrating protein expression with these endpoints would significantly strengthen the authors conclusions.

Minor comments:

1. There is ample negative data that, though important, can be largely relegated to supplement (e.g. SEA model/inflammation phenotype, cytokine models with neg inflammatory data).
2. Abstract. The authors use the term “induce asthma” loosely though mice do not get asthma and models only induce features found in asthma.
3. Page 3 – font is small for “Th2 cytokines”
4. Page 4. Tezepelumab is approved and has efficacy for both type 2 and non-type 2 asthma and should be mentioned.
5. Is there reason HDM was given IP instead of mucosal sensitization?
6. Figure 7. It would be interesting to identify whether IL-31 in combination with IL-13 leads to enhanced AHR suggesting a non-redundant role of IL-31 beyond IL-13.
7. Figure 6G/H. The labelling is confusing (IL-13 vs 31) with relation to the results stated in the text and the legend.
8. Figure 5A. IFN γ appears to be reduced in SEA treated IL-31 $^{-/-}$ mice. Could this be a mechanism for reduced AHR specifically in the SEA model? What is the explanation for this?
9. Of IFN γ and IL-13, which one is more potent to induce IL-31R expression?
10. Figure 8F. How does IFN γ induce FIZZ1 expression (STAT6-dependent gene)?
11. The immune cell phenotyping is limited in all models – what about ILCs, Th2 cells in IL-31R ko mice?
12. Figure 10. The changes in MLC2 phosphorylation in IL-31R KO cells seem modest compared to WT, it might be strengthened by showing either a dose or time-dependent effect with carbachol.

Reviewer #3 (expert in allergy, cytokines in asthma):

In this paper Akenepally et al examined the role of IL31 receptor A in airway hyperreactivity (AHR) and inflammation in a mouse model fo asthma. The authors showed that IL31RA $^{-/-}$ mice had attenuated AHR but not other features of asthma—inflammation and mucus production regardless of the inducing nature of the trigger—Type-1 or type-2. To address the mechanism, they examined the expression of

muscarinic acetylcholine receptors and found reduced expression of muscarinic acetylcholine receptor type-3 (CHRM3) and reduced phosphorylation of myosin light chain. The authors concluded that IL31RA functioned as a positive regulator of CHRM3 and calcium signaling in airway smooth muscle cells. The authors did an excellent job in demonstrating that IL31RA did not affect airway inflammation, mucus production and inflammatory cytokine production. They studied multiple models of asthma, which was another strength of the manuscript. However, the manuscript has many weaknesses. The major weakness is that it failed to provide a clear evidence for IL31RA regulation of airway smooth muscle CHRM3. The major concerns are outlined below.

1. The main conclusion that IL31RA regulates CHRM3 is not supported by appropriate experimental data. Many of the critical experiments (Fig 10) were performed in HEK293T cells overexpressing IL31RA and CHRM3, which is unacceptable. To prove their points, the authors need to do the following experiments: They need to demonstrate: 1) The expression of IL31RA and CHRM3 in airway smooth muscle from the mouse lung tissue; 2) Their interaction (co-precipitation studies) in airway smooth muscle cells; 3) IL31 induction of Ca²⁺ signaling in airway smooth muscle cells; 4) Absence of IL31-induced Ca²⁺ signaling in CHRM3^{-/-} smooth muscle cells.

2. Experiments presented in Figure 6 seem to go against the authors' conclusion. IL31 failed to induce airway hyperreactivity in vivo and ASMC-embedded collagen gel contraction in vitro. If IL31RA positively regulated airway smooth muscle Ca²⁺ signaling and muscle contraction, one would have expected induction of airway hyperreactivity and smooth muscle contraction by IL31.

Minor comments:

1. Fig. 2D: PAS staining is of poor quality. The arrowheads are misplaced.
2. Fig. 3, 5 and 8 show mRNA expression. The authors need to show data on protein expression.
3. Fig. 8: The authors show induction of mRNA for CCL11 by IFN γ . Did they see increased expression of the protein for this chemokine and eosinophilic influx in the lung? If eosinophils were not increased, then what was the relevance of increased CCL11 expression?
4. Fig. 10F: coprecipitation of IL31RA with CHRM3 from this overexpression model is difficult to interpret due to smearing of the CHRM3 band.
5. Is IL31RA expressed in human airway smooth muscle cells?

A point-by-point response to the reviewer's comments:

Reviewer 1

Summary and Novelty: Using two murine models of allergen-induced AHR, the authors determined that IL-31RA modulates baseline methacholine (Mch)-induced airway smooth muscle (ASM) shortening and calcium mobilization and attenuates allergen-induced AHR by decreasing ASM responsiveness that is independent of allergen-induced airway inflammation and goblet cell hyperplasia. The effects of IL-31RA on Mch responsiveness on ASM appear independent of IL-31 activation of IL-31RA but can be mediated by IL-4/IL-13- or IFN- γ -induced expression of IL-31RA on ASM that in turn enhances expression of ASM M3R expression and Mch-mediated calcium responses. The observation that IL-31RA modulates Mch-induced shortening, contraction, and calcium responses in murine ASM is novel. However, the recognition that allergen-induced AHR can be uncoupled from airway inflammation and goblet cell hyperplasia has been well established. The MS is well-written and the results support some of the conclusions; significant concerns should be addressed that would enhance the quality of the MS.

Overall Response: We appreciate the reviewer's supportive comments regarding the novelty of our findings. Also, we thank the reviewer for the additional comments that have helped us to improve the quality of the revised manuscript.

Major Comments

1) The findings exclusively using murine cells and models limit the physiological and pharmacological relevance to humans or to asthma. Human ASM cells are profoundly different from murine ASM; validation of the IL-31RA findings in human-derived ASM is critically important to understand the significance of their results.

Response: we agree with the reviewer's comment to validate our findings using human-derived ASM cells. Therefore, we used immortalized human ASMC (hTERT) that have been shown to retain CHRM3 expression and serve as a relevant cell model to assess M3 muscarinic signaling in several published studies (PMIDs: 17993586; 21205888; 34293268). We performed both the knockdown and overexpression of IL-31RA in hTERT cells and assessed changes in CHRM3 expression and carbachol-induced M3 muscarinic receptor signaling. The knockdown of IL-31RA attenuated CHRM3 expression (Fig. 9D & 9G), carbachol-induced calcium elevation (Fig. 10G), and MLC phosphorylation (Fig. 10H & Supplementary Fig. 11) in hTERT cells. Conversely, overexpression of IL-31RA augmented carbachol-induced calcium elevation in hTERT cells (Fig. 10C). Considering the lack of transcriptional regulation of CHRM3 expression by IL-31RA, we assessed cell surface levels of CHRM3 using biotinylation assay and show a positive regulation of CHRM3 by IL-31RA in both hTERT and HEK293 cells (Fig. 9D & Supplementary Fig. 10). Furthermore, we demonstrate the physical interaction between IL-31RA and CHRM3 in hTERT and HEK293 cells using proximity ligation assays (Fig. 9H and 9I). These new results are incorporated in the revised manuscript to support the proposed mechanism involving the IL-31AR-CHRM3 axis in ASMC contraction (Fig. 9 and 10).

2) Since non-specific AHR is a hallmark of asthma and since human ASM responds to many contractile agonists, the investigators should address whether their murine ASM findings are exclusively due to IL-31-M3R modulation. In other words, does IL-31 also modulate other agonist responses in murine ASM. The authors could use serotonin as an agonist; this question is critically important to understand how IL-31 RA mediates its effects.

Response: To address this question, we overexpressed IL-31RA in hTERT cells and treated these cells with multiple agonists including carbachol, bradykinin, and serotonin (Fig. 10C, 10D & 10E). Our new data on agonist-induced calcium elevation suggests that IL-31RA-driven effects are specific to carbachol, and do not modulate bradykinin or serotonin responses in both hTERT cells and HEK293 cells (Fig 10A -10G). In support, we observed no significant changes in the phosphorylation of MLC by serotonin in murine ASM cells (Supplementary Fig. 12). These new findings provide a more comprehensive understanding of the mechanisms underlying IL-31RA-driven ASM contractility. We included new data and modified text accordingly in the revised manuscript.

3) It is surprising that IL-31 alone had little effect on ASM excitation-contraction (EC) coupling. The global knockout of IL-31RA can have myriad effects that could contribute to the phenotype observed. This is a significant limitation of the studies. Does IL-31RA serve as a scaffold protein that enables G_q signaling? Such a possibility was not addressed.

Response: We appreciate the reviewer's comment regarding the surprising observation that IL-31RA alone had effects on ASM contractility. The knockdown and overexpression studies for IL-31RA using hTERT cells, HEK293 cells, and mouse ASMC suggest that IL-31RA functions as a positive regulator for CHRM3 expression and agonist-induced calcium elevation and MLC phosphorylation involved in smooth muscle cell contraction (Fig. 9 & 10). IL-31RA expression increased cell surface expression of CHRM3 yet had no effect on CHRM3 transcript suggesting a post-transcriptional mechanism involved in regulating CHRM3 expression. In support, studies using proximity ligation assays suggest a potential physical interaction between IL-31RA and CHRM3 in both hTERT and HEK293 cells (Fig. 9H and 9I). Importantly, the loss of function and gain of function studies suggest a positive regulation of the CHRM3 expression and CHRM3-mediated G_q signaling (calcium elevation and MLC phosphorylation) by IL-31RA in hTERT cells. While our studies suggest a potential chaperone function of IL-31RA for regulating cell surface expression of CHRM3, future studies are warranted to assess the additional mechanisms (e.g., scaffold for G_q activation) involved in the regulation of CHRM3 signaling and function by IL-31RA. We believe this is an important direction as suggested by the reviewer but beyond the scope of the current manuscript. We hope to communicate as a separate manuscript in the near future.

4. In the discussion, the investigators posit that targeting IL-31RA could mitigate MCh-induced AHR. How would the investigators modulate IL-31RA expression/activation since IL-31 alone has no effect? Certainly, the elimination of IL-31 would not be a viable approach and accordingly blockade of the receptor would not be useful.

Response: Our sincere apologies for the confusion with the above statement. Our new data using immunostaining demonstrate a significant increase in IL-31RA in SMC of airways in both mice and human lungs with asthma (Fig. 1D & 1E). Conversely, we observed no changes in the expression of IL-31 in HDM and SEA asthma models in vivo and had no effect on CHRM3 expression and contractility of ASMC (Fig. 1B-1E and 4B-4C) (Fig. 9B and 9C) (Fig. 5G & 5H). Therefore, we agree with the reviewer that neutralizing IL-31 or blocking IL-31RA may not be a viable option to mitigate the IL-31RA-CHRM3 axis in AHR. However, how IL-31RA-mediated regulation of AHR can be therapeutically exploited is not clear and requires additional studies. We now modified the discussion to reflect our new findings and the relevance of the IL-31RA-CHRM3 axis to mitigate AHR in asthma.

5. In Figure 1.D., the KD of IL-31RA at baseline attenuates Mch responses in the absence of HDM. Are these data significant compared to WT/saline? This is important and suggests that the global KO of IL-31 impairs ASM contractility. Indeed, does the KO of IL-31RA impair ALL smooth muscle responses to

activation of the M3R for example in the gut or vasculature? This is relevant if as the investigators posit, IL-31RA can be a therapeutic target.

Response: Thanks for the suggestion. We now updated our findings on AHR in saline-treated wildtype and IL-31RA knockout mice (Fig. 1D and 4D-4E). We observed a significant decrease in airway responsiveness in IL-31RA KO mice compared to wild-type mice treated with saline. Also, we performed PenH measurements to assess differences in airway reactivity at baseline and observed a significant decrease in lung resistance in mice deficient for IL-31RA compared to wildtype (Supplementary Fig. 3). To determine the relevance of IL-31RA in other SMC, we isolated SMC from the intestine of wildtype and IL-31RA knockout mice and assessed changes in carbachol-induced phosphorylation of MLC. Similar to ASMCS, we observed a significant decrease in the phosphorylation of MLC by carbachol with the loss of IL-31RA in intestinal SMC (Supplementary Fig. 13). Although our focus has been primarily on ASMC contractility in the context of asthma, our new findings from intestinal SMC suggest the broader impact of IL-31RA expression in SMC contractile responses in multiple organs. Overall, we believe our study findings would be valuable and open up future investigations to examine the effects of the IL-31RA-CHRM3 axis in SMC contractility and its potential as a therapeutic target beyond asthma.

Minor Comments

1. The title is inaccurate since the authors do not study human cells or tissue associated with asthma. The authors study allergen-induced AHR and airway inflammation.

Response: Based on the new data on immunostainings of IL-31RA in lung tissues and altered calcium signaling by the IL-31RA-CHRM3 axis in human ASMCS, we request the reviewer to consider our new title “Interleukin 31 receptor alpha regulates smooth muscle cell contraction and airway hyperresponsiveness in asthma”.

2. The introduction is too long and rambling; there should be more focus.

Response: The introduction is shortened with appropriate changes as suggested by the reviewer.

3. In the abstract, the authors state that both Th1 and Th2 cytokines induce asthma, which is incorrect. These cytokines are associated with asthma but there is no evidence of causation.

Response: As suggested, we now modified the sentence to “Th1 and Th2 cytokines are associated with asthma”

Reviewer 2

In the manuscript by Akkenepally et al., the authors show that loss of IL-31RA attenuated methacholine-induced AHR in HDM and SEA models, without affecting Th2 cytokine expression, mucus production, or recruitment of inflammatory immune cells. Interestingly, IL-31 administration did not have a significant effect on airway contractility. However, loss of IL-31RA attenuated IL-13-induced airway hyperreactivity without affecting IL-13's ability to induce goblet cell hyperplasia. Similarly, loss of IL-31RA attenuated IFN γ induced AHR without affecting inflammation or mucus production. IL-31RA deficiency led to a decrease in CHRM3 protein expression without changes in transcript level.

Furthermore, IL-31RA expression in HEK293T cells was able to augment carbachol-induced myosin light chain phosphorylation, calcium influx, possibly through interaction with CHRM3. Overall, this work identifies a novel mechanism through which IL-31RA selectively mediates AHR uncoupled from airway inflammation. However, there are specific issues that if addressed would significantly support the authors conclusions.

Overall Response: We thank the reviewer for valuable suggestions that have substantially helped us to improve the revised version of our manuscript.

Major Comments

1. Strain specificity. The authors use B6 mice strain and previous studies have shown that AHR differences can occur in strain-specific manners. To more broadly accept the conclusions regarding the important role of IL-31R selectively in AHR, IL-31/31R blocking studies in another strain of mice (e.g. Balb) in an HDM model would strengthen the overall work.

Response: We thank the reviewer for the suggestion to determine the strain-specific differences in IL-31RA-driven AHR. In the current study, we demonstrated the effects of IL-31RA deficiency on AHR using two alternative mouse models of allergic asthma and treatment with exogenous administration of IL-13 and IFN- γ , albeit using one strain. While establishing strain differences is valuable, repeating these studies in another strain is beyond the scope of the manuscript. Most of our additional efforts were dedicated to establishing the mechanism involved in IL-31RA-induced upregulation of CHRM3-mediated ASMC contraction.

2. The current manuscript demonstrates that IL-31R is important for the development of AHR but a critical question therapeutically is whether blockade of IL-31/31R after lung inflammation is established will also show the same reduction in AHR. IL-31/R blocking studies could address this,.

Response: We thank the reviewer for the suggestion of exploring IL-31RA blocking antibodies. While we agree with the reviewer that blocking IL-31RA after lung inflammation is established would be an exciting new idea, such studies are not feasible at this point in time considering the lack of murine IL-31RA antibody. Generating an effective antibody for animal studies time consuming, expensive, and cost prohibitory. We have highlighted this possibility in the Discussion as a future direction.

3. The in vitro studies showing a potential interaction between CHRM3 and IL-31R to regulate ASM calcium influx and MLC phosphorylation are interesting. Gain of function studies in HEK293T cells should be complemented by loss of function through siRNA knockdown or genetic knockout of CHRM3 to see if this attenuates IL-31RA intracellular calcium flux. Further, less artificial studies in ASM after Th2 cytokine upregulation of IL-31R to assess CHRM3-IL-31R interactions would be more relevant to the in vivo data.

Response: The gain-of-function studies suggest that CHRM3 overexpression increases calcium influx and this increase is further augmented with the co-expression of IL-31RA (Fig 10A). We agree with the reviewer that the gain of function studies should be complemented by the loss of function studies. Accordingly, we now performed the IL-31RA knockdown and assessed carbachol-driven calcium release and MLC phosphorylation in hTERT cells. Consistent with our overexpression studies, we observed a significant attenuation in carbachol-induced calcium elevation and phosphorylation of MLC in hTERT cells with the loss of IL-31RA (Fig. 10G & 10H). Also, our new findings using proximity ligation studies demonstrate the physical binding interactions between IL-31RA and CHRM3 in hTERT cells (Fig. 9H and 9I). We believe these new findings are novel and critical to accelerate follow-up studies in assessing mechanisms underlying Th2 cytokine regulation of the IL-31RA and CHRM3 interactions in ASMC contraction. Currently, studies are ongoing to assess the effects of both Th1 and Th2 cytokines either alone or in combination in modulating IL-31RA expression, membrane

colocalization, and CHRM3-mediated calcium elevation and MLC phosphorylation. It would be difficult to accommodate these new findings in the current manuscript but will be communicated as a separate manuscript.

4. The manuscript shows mostly transcript levels of key endpoints (cytokines, IL-31R expression) without corroborative protein expression. Further, tissue protein staining of IL-31R at baseline and after T2 HDM model could support the more selective role of IL-31R in ASM function compared with function in other cell types. Overall, demonstrating protein expression with these endpoints would significantly strengthen the authors conclusions.

Response: We agree and have now performed immunostaining for IL-31RA in mice sensitized and challenged with HDM and human lungs obtained from donors with asthma. Our immunostaining studies show the detection of IL-31RA selectively in SMC of both mice and human lungs and no staining with isotype control antibodies (Supplementary Fig. 1). Consistent with our RT-PCR data, we observed heightened immunostaining for IL-31RA in HDM-challenged mice compared to saline-treated mice and in lungs obtained from donors with asthma compared to healthy donors (Fig. 1D and 1E). Further, we measured cytokine protein levels in BALF of wildtype and IL-31RA knockout mice treated with saline and HDM. Consistent with the lung transcript levels, we observed elevated levels of IL-4, IL-13, IL-5, IL-6, IFN γ , TNF α , and IL-10 in HDM-challenged compared to saline-treated wildtype and IL-31RA knockout mice (Fig. 3E). We have updated our manuscript with the newly generated protein expression data and revised the text to highlight these findings and their implications in the context of our study.

Minor Comments

1. There is ample negative data that, though important, can be largely relegated to supplement (e.g. SEA model/inflammation phenotype, cytokine models with neg inflammatory data).

Response: We have moved large amounts of RT-PCR data on cytokine gene expression, and data from SEA model studies to supplement.

2. Abstract. The authors use the term “induce asthma” loosely though mice do not get asthma and models only induce features found in asthma.

Response: Suggested change is now included in the revised manuscript

3. Page 3 – font is small for “Th2 cytokines”

Response: Suggested change is now included in the revised manuscript

4. Page 4. Tezepelumab is approved and has efficacy for both type 2 and non-type 2 asthma and should be mentioned.

Response: Suggested change is now included in the revised manuscript

5. Is there reason HDM was given IP instead of mucosal sensitization?

Response: The previously published studies from our group and others showed robust Th2 cytokine responses, AHR, goblet cell hyperplasia, and inflammation with the IP sensitization model. Also, our recently published study showed elevated expression of IL-31RA in wildtype mice challenged with SEA via IP and this increase attenuated with the loss of IL-13RA1 expression.

6. Figure 7. It would be interesting to identify whether IL-31 in combination with IL-13 leads to enhanced AHR suggesting a non-redundant role of IL-31 beyond IL-13.

Response: We observed no significant changes in airway contractility or SMC contraction by IL-31 in PCLS and ASMC. Therefore, we anticipate no significant changes with the addition of IL-31 in IL-13-driven SMC contractility. However, we will test this combination as well as other Th1 and Th2 cytokines in future studies as suggested by Reviewer 2 (see Reviewer 2 and Response 3).

7. Figure 6G/H. The labelling is confusing (IL-13 vs 31) with relation to the results stated in the text and the legend. Updated in the figure

Response: Suggested change is now included in the revised manuscript.

8. Figure 5A. IFN γ appears to be reduced in SEA treated IL-31 $^{-/-}$ mice. Could this be a mechanism for reduced AHR specifically in the SEA model? What is the explanation for this?

Response: It is difficult to conclude that reduced IFN γ is a mechanism for reduced AHR based on modest transcriptional changes for IFN γ in the SEA model. However, we measured the protein levels of IFN γ in the BALF and observed no significant changes with the loss of IL-31RA in saline or HDM-challenged mice (Fig. 3E). To avoid confusion, we now removed the data on IFN γ transcript levels in the SEA model and other transcripts measured in the SEA model are now included as supplementary fig. 5.

9. Of IFN γ and IL-13, which one is more potent to induce IL-31R expression?

Response: We observed a dose-dependent increase in IL-31RA expression in ASMC treated with both IL-13 and IFN γ . It may be not possible to assess the differences in the potency between IL-13 and IFN γ in inducing IL-31RA expression using RT-PCR conclusively.

10. Figure 8F. How does IFN γ induce FIZZ1 expression (STAT6-dependent gene)?

Response: We don't know the mechanism underlying increased FIZZ1 transcripts in the lungs of IFN γ -treated mice. Also, the observed changes in FIZZ1 transcripts may not be relevant to the scope of the current manuscript and thus data was removed in the revised version of the manuscript.

11. The immune cell phenotyping is limited in all models – what about ILCs, Th2 cells in IL-31R ko mice?

Response: Due to limited or no changes in inflammation or cytokines associated with ILCs and Th2/Th1 cytokines with the loss of IL-31RA, we have not performed detailed immune cell phenotyping. Also, we focused more on SMC as we observed robust changes in the contractility and calcium signaling with the loss of IL-31RA in ASMC.

12. Figure 10. The changes in MLC2 phosphorylation in IL-31R KO cells seem modest compared to WT, it might be strengthened by showing either a dose or time-dependent effect with carbachol.

time dependent cch treatment will be done for 0,5,15,30 min in both wildtype and il-31RA ko

Response: Additional time points to assess the time-dependent effects of carbachol in the phosphorylation of MLC20 using hTERT cells are now included (Fig 10H and Supplementary Fig. 11).

Reviewer 3

In this paper Akenneppally et al examined the role of IL31 receptor A in airway hyperreactivity (AHR) and inflammation in a mouse model for asthma. The authors showed that IL31RA $^{-/-}$ mice had attenuated AHR but not other features of asthma—inflammation and mucus production regardless of the inducing nature of the trigger—Type-1 or type-2. To address the mechanism, they examined the expression of muscarinic acetylcholine receptors and found reduced expression of muscarinic acetylcholine receptor type-3 (CHRM3) and reduced phosphorylation of myosin light chain. The authors concluded that IL31RA functioned as a positive regulator of CHRM3 and calcium signaling in airway smooth muscle cells. The authors did an excellent job in demonstrating that IL31RA did not affect airway inflammation, mucus production and inflammatory cytokine production. They studied multiple models of asthma, which was another strength of the manuscript. However, the manuscript has many weaknesses. The major weakness is that its failed to provide a clear evidence for IL31RA regulation of airway smooth muscle CHRM3.

Response: We thank the reviewer for the valuable feedback and suggestions. Below, we have provided detailed responses to each point raised.

The major concerns are outlined below.

1. The main conclusion that IL31RA regulates CHRM3 is not supported by appropriate experimental data. Many of the critical experiments (Fig 10) were performed in HEK293T cells overexpressing IL31RA and CHRM3, which is unacceptable. To prove their points, the authors need to do the following experiments: They need to demonstrate: 1) The expression of IL31RA and CHRM3 in airway smooth muscle from the mouse lung tissue; 2) Their interaction (co-precipitation studies) in airway smooth muscle cells; 3) IL31 induction of Ca²⁺ signaling in airway smooth muscle cells; 4) Absence of IL31-induced Ca²⁺ signaling in CHRM3^{-/-} smooth muscle cells.

Response: We would like to thank the reviewer for the constructive comments, and we have carefully considered each of your suggestions and have made the necessary revisions to address all your concerns.

i) we performed immunostainings to determine the expression levels of IL31RA in the airways of HDM-challenged mouse lungs. The results, which are presented in Fig. 1D demonstrate increased staining for IL-31RA in SMC of airways, thus confirming their physiological relevance. Further, we observed elevated immunostaining for IL-31RA in SMC of asthmatic human airways compared to healthy controls (Fig. 1E).

ii) To investigate the interaction between IL31RA and CHRM3 in human ASMC, we performed proximity ligation studies using hTERT cells. (Fig. 9H and 9I).

iii) We have conducted experiments to examine the effect of IL-31RA on Ca²⁺ signaling in hTERT cells. The results, depicted in Fig. 10A-10F, illustrate that IL-31RA overexpression leads to a significant increase in intracellular Ca²⁺ levels. Similarly, the loss of IL-31RA decreased intracellular Ca²⁺ levels (Fig. 10G). These findings suggest IL-31RA-dependent upregulation of CHRM3-mediated calcium influx. Similarly, we assessed carbachol-induced phosphorylation of MLC in ASM cells.

In summary, the signaling studies are performed in human ASM (hTERT) cells, primary murine ASM cells (wild-type and IL-31RA knockout mice), and HEK293 cells. Both gain and loss of function approaches are used wherever appropriate.

2. Experiments presented in Figure 6 seem to go against the authors' conclusion. IL31 failed to induce airway hyperreactivity in vivo and ASMC-embedded collagen gel contraction in vitro. If IL31RA positively regulated airway smooth muscle Ca²⁺ signaling and muscle contraction, one would have expected induction of airway hyperreactivity and smooth muscle contraction by IL31.

Response: The experiments depicted in Fig. 5 in the revised manuscript were designed to investigate the effects of IL31 on airway hyperreactivity and ASMC-embedded collagen gel contraction. We agree that the results from these experiments may appear contradictory to our conclusion regarding the positive regulation of Ca²⁺ signaling and contraction of ASMC by IL31RA. However, we believe there are several important factors to consider in interpreting these results. Firstly, it is possible that IL-31RA-driven effects on AHR and contraction of ASMC could be independent of its cognate ligand, IL-31. Based on the findings in Fig. 5, we primarily focused on the effects of IL31RA on CHRM3 expression and Ca²⁺ signaling and MLC phosphorylation in ASMC. Notably, our new findings using cell surface biotinylation studies suggest that cell surface expression of CHRM3 is dependent on IL-31RA in both hTERT and HEK293 cells (Fig. 9D and Supplementary Fig. 10). Secondly, it is possible that other unknown ligands bind and modulate the effects of IL-31RA. Further, IL-31RA-mediated regulation of

CHRM3-mediated smooth muscle contraction extends beyond airways as we observed similar findings in intestinal smooth muscle cells. Therefore, the lack of IL-31 effects on AHR and contraction of SMC may reflect the multifaceted nature of these processes rather than contradicting the role of IL-31RA in CHRM3-driven Ca^{2+} signaling. Future studies will assess the ligand-independent effects of IL-31RA in stabilizing and trafficking of CHRM3 in ASMC.

1. Fig. 2D: PAS staining is of poor quality. The arrowheads are misplaced.

Response: New images are now included in the revised manuscript and arrowheads are adjusted.

2. Fig. 3, 5 and 8 show mRNA expression. The authors need to show data on protein expression.

Response: Suggested change is now included in the revised manuscript (Fig. 3E).

3. Fig. 8: The authors show induction of mRNA for CCL11 by IFN γ . Did they see increased expression of the protein for this chemokine and eosinophilic influx in the lung? If eosinophils were not increased, then what was the relevance of increased CCL11 expression?

Response: We measured mRNA for CCL11 but not protein. H&E staining suggests limited or no changes in eosinophils. Further, our ongoing studies using ASMC treated with IFN γ suggest an increased contraction of ASMC that could be independent of eosinophils or CCL11.

4. Fig. 10F: coprecipitation of IL31RA with CHRM3 from this overexpression model is difficult to interpret due to smearing of the CHRM3 band.

Response: This could be due to the strong interaction between IL-31RA and CHRM3. Alternatively, we assessed and show this interaction using a proximity ligation assay in both HEK293 cells and hTERT cells (Fig. 9H and 9I).

5. Is IL31RA expressed in human airway smooth muscle cells?

Response: Our new findings using hTERT cells and immunostainings on lung tissues demonstrate IL-31RA expression in both mouse and human SMC (Fig. 1 D, 1E, 9D, 9G, and 9I).

REVIEWERS' COMMENTS

Reviewer #1 (expert in airway hyperresponsiveness and airway smooth muscle cells):

Interleukin 31 receptor alpha regulates smooth muscle cell contraction and airway hyperresponsiveness in asthma.

Akkenepally S., et. al.

Major concerns:

1. I appreciate the responsiveness of the authors to my queries. New experiments were added to show IL-31RA expression airway smooth muscle (ASM) in asthma; ASM levels of IL31RA expression were greater when compared to that from non-asthma donors. Although the immunohistochemical data is striking, the reader has no idea whether this occurs in all asthma patients or the correlation with the severity of disease or concomitant drug use. This should be addressed.
2. The overall magnitude of the IL31RA effects on Mch responsiveness is modest as shown in Figure 1f. There exists a 50% dependence of Mch responsiveness on IL-31RA in the allergen-induced sensitization/challenge models (Fig. 1f). There is also a 50% dependence of Mch responsiveness on IL-13RA at baseline. Importantly, Mch responsiveness of ASM does not exclusively require IL-31RA since ASM still responds to Mch despite a lack of IL-31RA. This point was not addressed or identified. The mechanism by which IL-31RA modulates M3R responses remains unclear.
3. The data in hTERT human ASM cells and HEK293 cells support the authors hypotheses but is quite contrived since HEK293 cells don't natively express CHRM3. The new experiments that show overexpression of IL-31RA in ASM selectively increases Mch responses is consistent with the authors hypotheses.
4. The observation that airway inflammation in asthma or in allergen sensitization/challenge models is uncoupled from AHR is not novel. Although the new data support some aspects of the authors hypotheses, the lack of a mechanism by which IL-31RA modulates CHRM3 excitation-coupling and the magnitude of the response lessens enthusiasm.

Minor concerns: none

Reviewer #2 (expert in asthma and lung inflammation):

The authors have significantly improved the mechanistic studies in the revised manuscript and adequately responded to my comments.

Reviewer #3 (expert in allergy, cytokines in asthma):

The authors have satisfactorily responded to my critique. They performed new experiments to address my concerns. Despite their earnest effort they were unable to demonstrate a direct effect of IL31 on airway hyperactivity and calcium signaling. The authors postulate a ligand-independent mechanism or the existence of an alternative ligand for IL31 receptor. The effect of IL31RA on muscarinic acetylcholine receptor 3 in airway smooth muscle is novel and is likely to have functional implications. I recommend the acceptance of the revised manuscript.

A point-by-point response to the reviewer's comments:

Reviewer #1

Major concerns:

1. I appreciate the responsiveness of the authors to my queries. New experiments were added to show IL-31RA expression airway smooth muscle (ASM) in asthma; ASM levels of IL31RA expression were greater when compared to that from non-asthma donors. Although the immunohistochemical data is striking, the reader has no idea whether this occurs in all asthma patients or the correlation with the severity of disease or concomitant drug use. This should be addressed.

Response: We would like to thank the reviewer for acknowledging our responsiveness and the new experimental data that supports a unique role for the IL-31RA-CHRM3 axis in ASMC contraction and AHR in asthma. We also thank the reviewer for agreeing with our interpretation of the upregulation of IL-31RA expression in the ASMC of asthmatic airways in both human and mouse lung tissues. These findings are significant and justify further investigations into the role of IL-31RA in asthma pathogenesis. However, we are of the opinion that utilizing histological staining may not be a reliable and feasible method for assessing the association between IL-31RA levels and the severity of asthma or drug usage. This is especially due to the temporal and spatial variations in lung tissues collected and heterogeneity in asthma. Furthermore, such an assessment falls beyond the scope of the current study. We have alluded to this point in the discussion (Page 20).

2. The overall magnitude of the IL31RA effects on Mch responsiveness is modest as shown in Figure 1f. There exists a 50% dependence of Mch responsiveness on IL-31RA in the allergen-induced sensitization/challenge models (Fig. 1f). There is also a 50% dependence of Mch responsiveness on IL-13RA at baseline. Importantly, Mch responsiveness of ASM does not exclusively require IL-31RA since ASM still responds to Mch despite a lack of IL-31RA. This point was not addressed or identified. The mechanism by which IL-31RA modulates M3R responses remains unclear.

Response: We agree with the reviewer's comment regarding the 50% dependence on MCh responsiveness by IL-31RA in both naïve and allergen-challenged mice. In vivo responsiveness of airways is quite complex and entails the engagement of M2 muscarinic receptors in addition to the M3 receptor. The reductionist approaches using cell-based studies are valuable in ascertaining the regulation of the M3 signaling/function by IL-31RA. Using the knockdown and overexpression of IL-31RA, we have established that IL-31RA regulates the M3 expression and signaling. Our mechanistic studies reported in the manuscript have ruled out the transcriptional regulation of the M3 by IL-31RA and suggested posttranscriptional regulation mechanism(s). We believe that these new findings serve as the groundwork for future studies aimed at identifying additional (posttranscriptional) mechanisms through which IL-31RA modulates the M3 expression and signaling. We have included additional discussion on M2 vs. M3 muscarinic receptors in the lung contributing to airway responsiveness (Page 22).

3. The data in hTERT human ASM cells and HEK293 cells support the authors hypotheses but is quite contrived since HEK293 cells don't natively express CHRM3. The new experiments that show overexpression of IL-31RA in ASM selectively increases Mch responses is consistent with the authors hypotheses.

Response: We apologize for any confusion that may have arisen. The previously published (PMID: 11036069) study has demonstrated endogenous expression of M3 in HEK293 cells, and these findings align with our results obtained using HEK293 cells. We genuinely appreciate the reviewer's comment

that our findings are consistent and support the hypothesis that the IL-31RA promotes M3 levels and subsequent calcium signaling in ASMC.

4. The observation that airway inflammation in asthma or in allergen sensitization/challenge models is uncoupled from AHR is not novel. Although the new data support some aspects of the authors hypotheses, the lack of a mechanism by which IL-31RA modulates CHRM3 excitation-coupling and the magnitude of the response lessens enthusiasm.

Response: We agree with the reviewer's comment that uncoupling between AHR and inflammation is not a new concept in asthma. However, identifying the role of IL-31RA in uncoupling AHR from inflammation is novel and likely important, as noted by reviewer 3. We believe that the current findings serve as a foundation for future studies aimed at developing new therapies to mitigate the effects of IL-31RA on altered M3 expression and signaling in ASMC.

Minor concerns: none

Reviewer #2 (expert in asthma and lung inflammation):

The authors have significantly improved the mechanistic studies in the revised manuscript and adequately responded to my comments.

Response: We are thankful to the reviewer for the critical comments in achieving completeness and improving the quality of the manuscript, a necessary step for publication in Nature Communications.

Reviewer #3 (expert in allergy, cytokines in asthma):

The authors have satisfactorily responded to my critique. They performed new experiments to address my concerns. Despite their earnest effort they were unable to demonstrate a direct effect of IL31 on airway hyperactivity and calcium signaling. The authors postulate a ligand-independent mechanism or the existence of an alternative ligand for IL31 receptor. The effect of IL31RA on muscarinic acetylcholine receptor 3 in airway smooth muscle is novel and is likely to have functional implications. I recommend the acceptance of the revised manuscript.

Response: We are thankful to the reviewer for supportive comments on the novelty of current findings and suggestions, which greatly contributed to improving the manuscript.